https://doi.org/10.1038/s41467-021-25031-6　　**OPEN**

# Shear band-driven precipitate dispersion for ultrastrong ductile medium-entropy alloys

Tae Jin Jang[1,6], Won Seok Choi[2,6], Dae Woong Kim[3], Gwanghyo Choi[2], Hosun Jun [2], Alberto Ferrari [4], Fritz Körmann[4,5], Pyuck-Pa Choi [2✉] & Seok Su Sohn [1✉]

Precipitation strengthening has been the basis of physical metallurgy since more than 100 years owing to its excellent strengthening effects. This approach generally employs coherent and nano-sized precipitates, as incoherent precipitates energetically become coarse due to their incompatibility with matrix and provide a negligible strengthening effect or even cause brittleness. Here we propose a shear band-driven dispersion of nano-sized and semicoherent precipitates, which show significant strengthening effects. We add aluminum to a model CoNiV medium-entropy alloy with a face-centered cubic structure to form the L2$_1$ Heusler phase with an ordered body-centered cubic structure, as predicted by ab initio calculations. Micro-shear bands act as heterogeneous nucleation sites and generate finely dispersed intragranular precipitates with a semicoherent interface, which leads to a remarkable strength-ductility balance. This work suggests that the structurally dissimilar precipitates, which are generally avoided in conventional alloys, can be a useful design concept in developing high-strength ductile structural materials.

[1] Department of Materials Science and Engineering Korea University, Seoul, South Korea. [2] Department of Materials Science and Engineering Korea Advanced Institute of Science and Technology, Daejeon, South Korea. [3] Center for High Entropy Alloys Pohang University of Science and Technology, Pohang, South Korea. [4] Department of Materials Science and Engineering Delft University of Technology, Mekelweg 2, Delft, The Netherlands. [5] Max-Planck-Institut für Eisenforschung Max-Planck-Straße 1, Düsseldorf, Germany. [6] These authors contributed equally: Tae Jin Jang and Won Seok Choi. ✉email: p.choi@kaist.ac.kr; sssohn@korea.ac.kr

Striving for developing materials with ultrahigh strengths and sufficient uniform ductility has been an imperative challenge in structural applications. The aim has been usually achieved by configuring a disordered matrix responsible for ductility with ordered phases as precipitates to hinder dislocation motion. This so-called precipitation strengthening, especially when introducing structurally similar second phases coherent to the matrix, is remarkably effective in restraining the sacrifice of ductility[1–3]. The coherency enables the precipitates to distribute homogeneously in nanoscale, resulting in coherency strain fields and antiphase-boundary strengthening, as for instance in multicomponent Co-based or Ni-based superalloys consisting of a $L1_2$ ordered face-centered-cubic (FCC) phase in a disordered FCC matrix[4]. However, excessive alloying of the ordered phase-forming elements may lead to the formation of phases structurally dissimilar to the matrix called topologically closed-packed (TCP) phases such as the σ phase, the μ phase, and the Laves phases, resulting in the inhomogeneous distribution at grain boundaries at a coarse microscale due to a loss of coherency[5–7]. This microstructure makes the alloys prone to catastrophic failure in load-bearing applications. Consequently, the incoherent precipitates have been perceived as detrimental, to be suppressed by delicate controls of alloying elements and thermomechanical treatments[8–10].

Nevertheless, to conquer the severe brittleness of dissimilar phases with a matrix, one feasible way is to manipulate semicoherent precipitates by avoiding the formation of incoherent interfaces with the matrix. The lower interfacial energy of semicoherent precipitates than incoherent ones reduces their coarsening, while the homogeneous distribution in nanoscale can be embodied by introducing additional nucleation sites in the lattice, as reported in low-density steel and Al alloys[11,12]. To implement our design philosophy and develop ultrastrong alloys with good ductility, we choose an equiatomic ternary Co–Ni–V alloy as a model matrix system. This medium-entropy alloy (MEA), as a subclass of multiprincipal element alloys (MPEAs) or high-entropy alloys (HEAs) possessing single-phase structure[13–15], exhibits great mechanical properties, in particular a yield strength of ~1 GPa, attributed to severe lattice distortion, and a tensile ductility of 38%[15]. This property is ascribed from only solid-solution and grain-boundary strengthening at an average ~2 μm grain size in FCC-structured matrix. However, it is challenging to further enhance the mechanical properties of CoNiV alloys as an additional refinement of grains is restricted in conventional processing due to the limited process windows[16]. In this respect, precipitation strengthening can be an attractive candidate for further improving the mechanical properties. Recently, unceasing endeavors have been exerted in developing MPEAs through precipitation strengthening, and several studies demonstrated that homogenously distributed $L1_2$ nanoparticles in a FCC matrix are particularly effective in significantly enhancing strength while retaining moderate ductility[17–22]. Most of the studies exploited coherent precipitates, but there was no attempt yet to achieve improved properties by adopting semicoherent precipitates dissimilar to the matrix[23–25].

Here we present a CoNiV-based MEA that can be strengthened through the formation of semicoherent nanoprecipitates and thermomechanical treatments enabling them to disperse homogeneously in the lattice. To realize such a material, ~6.25 at% Al is added to form an $L2_1$ ordered body-centered-cubic (BCC) phase in a FCC matrix, based on density-functional theory (DFT) calculations. A conventional cold-rolling process is conducted to introduce sufficient lattice defects into the material, and a subsequent heat treatment promoted the formation of precipitates and the recrystallization, assisted by the stored energy near the lattice defects. The material processes tailoring the size and morphology of the precipitates, with the aid of the high dislocation density, allow for effective strengthening, leading to a remarkable strength–ductility balance. Our approach demonstrates that structurally dissimilar precipitates, which are generally avoided due to their negligible strengthening or detrimental effect on ductility, can provide a useful design concept for the development of high-strength ductile structural materials.

## Results

**Material and process design.** Aluminum is one of the most widely used elements in HEAs and MEAs as secondary phase former. The ordered phases that can be generated by the sole addition of Al or combined with Ti are the B2 (CoAl, NiAl) and $L1_2$ ($Co_3Al$, $Ni_3Al$) phases for Co–Ni-containing FCC solid-solution alloys, such as CoCrNi, CoCrFeNi, CoCrCuFeNi, or CoCrFeMnNi[19,26–29]. In this study, we added Al to the CoNiV MEA, which showed enhanced mechanical properties imputed to severe lattice distortion[15]. DFT calculations (Fig. 1a) determined the most stable precipitate as a Co-rich $L2_1$ phase among the ordered $L1_2$-$(Co,Ni)_3Al$, B2-$(Co,Ni)Al$, and $L2_1$-$(Co,Ni)_2VAl$ phases, indicating that the high concentration of V affects the stability of ordered phases. The stabilization of this phase originates from the opening of a pseudogap in the electronic density of states of the $L2_1$ phase (Fig. 1b) in proximity to the Fermi level ($E_F$), which is a common indication of a compound with high formation energy because more electronic states can be accommodated at lower energies. As for the stoichiometry of this $L2_1$ phase, we note that the larger the Co concentration, the closer the pseudogap to $E_F$, signaling increased stability of the $L2_1$ phase with an increasing amount of Co in this precipitate (see Supplementary Note for detailed information).

Figure 1c displays the overall schematics of the thermomechanical processing to fabricate the precipitation-strengthened $Al_{0.2}CoNiV$ alloys. As predicted, the homogenized state consisted of a coarse FCC matrix and Al-rich $L2_1$ islands (Supplementary Fig. 1), in which the cold-rolling process-induced macroscopic and microscopic shear bands (Supplementary Fig. 2). The macroshear bands were formed during cold-rolling and consequent strain localization, resulting in thick lines distributed at ~10–45° to the rolling direction. These macroshear bands do not need to follow crystallographic orientations. On the other hand, micro-shear bands were developed inside the FCC grains alongside the macroshear bands. It is known that the micro-shear bands can be in the form of twins bundles, dense stacking-fault bundles, or linear arrays of dislocations and thus they are aligned along specific crystallographic shear planes[30]. After subsequent recrystallization and aging processes (1150 °C for 2 h and 850 °C for 1 h, respectively), referred to as RA, the $L2_1$ phase was also observed in the form of islands and particles at grain boundaries or triple junctions (Supplementary Fig. 3). These $L2_1$ precipitates at grain boundaries do not significantly contribute to strengthening because it is difficult for dislocations to directly interact with precipitates during glides, but they rather readily pile up in front of precipitates at the pre-existing pile-up sites, i.e., the grain boundaries. Instead, here we utilized the abundant micro-shear bands as nucleation sites via partial recrystallization treatment for obtaining nanosized precipitates finely dispersed inside grains. Therefore, the heat-treatment conditions were selected to be 800, 850, and 900 °C for 1 h to control the recrystallization and precipitation behavior.

**Microstructural characterization.** The MEA designed in this work (V: 31.25, Co: 31.25, Ni: 31.25, Al: 6.25 (in at%) or $Al_{0.2}CoNiV$), heat-treated at three different temperatures, are hereafter referred to as A800, A850, and A900, respectively (see

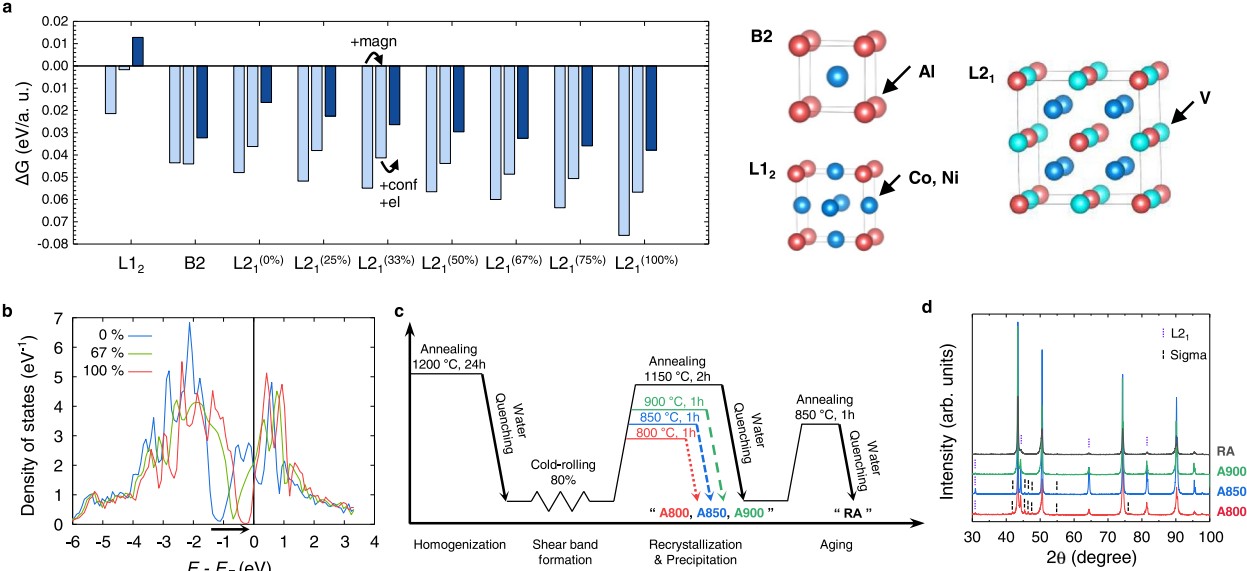

**Fig. 1 Material and process design. a** Free energy of the candidate precipitates with respect to the solid solution. The percentages for the L2$_1$ phase indicate the occupation of the (Co,Ni) sublattice by Co. For each phase, the first, second, and third bars represent the 0 K (−273 °C) energy difference in the ferromagnetic state, the 0 K (−273 °C) energy difference in the paramagnetic state, and the free energy difference at 1150 K (877 °C) with the inclusion of configurational and electronic entropy, respectively. **b** Densities of states of the L2$_1$ phase for different Co contents in the ferromagnetic state. **c** Schematics of the thermomechanical processing for precipitation-strengthened Al$_{0.2}$CoNiV alloys. **d** Phase identification via X-ray diffraction (XRD) analysis of the annealed Al$_{0.2}$CoNiV alloys.

"Methods"). As shown by X-ray diffraction (XRD) (Fig. 1d), the primary peaks of the alloys corresponded to the FCC phase, while the secondary peaks were superlattice reflections stemming from the L2$_1$-ordered phase. The A850 and A800 alloys additionally contained the σ phase, present for the most part in the L2$_1$ islands and with a smaller fraction also in the nearby recrystallized FCC grains (Supplementary Fig. 4). Figure 2a–f shows electron back-scatter diffraction (EBSD) maps for the investigated alloys, revealing FCC (Fig. 2a–c) and L2$_1$ (Fig. 2d–f) phases. For A800 (Fig. 2a), the FCC grains show a bimodal size distribution resulting from partial recrystallization. The recrystallized FCC grains have an average size of ~1 μm, and are distributed along macroshear bands formed at ~45° to the cold-rolling direction. This bimodal distribution can be attributed to the high stored energy in the shear bands, which promote heterogeneous nucleation. The coarse FCC grains presented in the as-homogenized state are refined by macroscopic shear bands induced from cold-rolling, leading to an average grain size of 212.1 ± 73.4 μm. The L2$_1$ grains also show a bimodal size distribution, with a size of 7.4 μm for the coarse island and a few hundred nanometers for the fine particles (Fig. 2d). Electron-channeling contrast imaging (ECCI) shows the microstructure of the present alloys more precisely (Fig. 2g–i). For A800, the L2$_1$ particles exhibit a fine rod-like shape with an average diameter of ~57 nm and aspect ratio of ~3, and they are aligned along the micro-shear bands inside the non-recrystallized FCC matrix. These L2$_1$ nanoparticles have Kurdjumov–Sachs (K–S) orientation relationship with the FCC matrix, which will be discussed later in this report. It is noteworthy that very fine additional FCC laths are observed inside the L2$_1$ islands, having a width of ~40 nm and interspacing of ~29 nm.

As the heat-treatment temperature increases, the fraction and size of the recrystallized FCC grains increase, resulting in a completely recrystallized structure in the A900 alloy (Fig. 2a–c). L2$_1$ precipitates start to form at the grain boundaries or triple junctions with recrystallization process. These intergranular L2$_1$

particles exhibit an incoherent or only one-sided semicoherent interface with the adjacent disordered FCC grains (Supplementary Fig. 5). For the fully recrystallized sample (A900), the L2$_1$ particles at grain boundaries or triple junctions in the recrystallized FCC region are ~320 nm in size. The L2$_1$ precipitates are also present inside the recrystallized FCC grains as intragranular particles. These intragranular L2$_1$ particles show an average size of ~146 nm. The FCC laths grow to ~250 nm and the σ phase disappears. The fractions of the constituent phases and their sizes are summarized in Supplementary Tables 1 and 2, respectively. Figure 2j, k displays the overall microstructural evolutions of Al$_{0.2}$CoNiV alloys.

Figure 3a–d shows a detailed EBSD analysis of the A800 sample. The recrystallized FCC grains are located at the interface between the non-recrystallized FCC and L2$_1$ islands. This preferred nucleation behavior of FCC grains can be attributed to the strain incompatibility between the soft FCC phase and the hard L2$_1$ island, which can be confirmed by the high kernel average misorientation (KAM) values at their interfaces (Fig. 3b). Compared to the dislocation-free recrystallized grains, the non-recrystallized FCC grains show evident green lines along the {111} plane trace indicating a high dislocation density ($1.77 \times 10^{15}$ m$^{-2}$) of non-recrystallized grains. This result demonstrates that the cold-rolling leads to the micro-shear bands along specific crystallographic {111} planes as well as to macroshear bands. The crystallographic orientation of FCC laths is almost identical in each L2$_1$ island, implying that the two phases have well-defined orientation relationships (Supplementary Fig. 5).

The role of micro-shear bands on the L2$_1$ formation is revealed in Fig. 3e–h. An ECCI map shows that the L2$_1$ particles are well aligned along the micro-shear bands in a non-recrystallized FCC grain of the A800 alloy (Fig. 3e). Notably, distinctively contrasting line features along the micro-shear band are apparent, as indicated by the arrows. A transmission electron microscopy (TEM) image (Fig. 3f–h) shows such a precipitate region and an

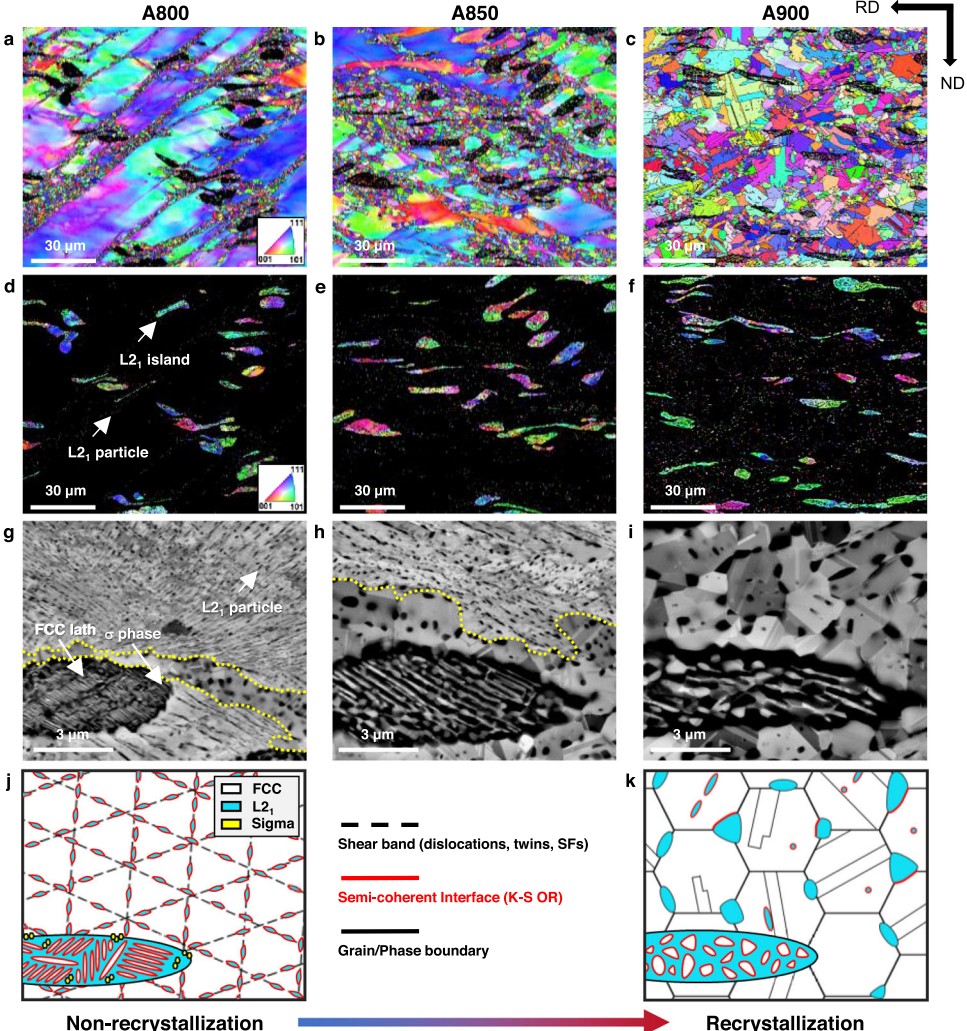

**Fig. 2 Microstructural evolutions with increased heat-treatment temperature. a–c** Electron backscatter diffraction (EBSD) maps of face-centered-cubic (FCC), **d–f** EBSD maps of L2$_1$ phases, **g–i** electron-channeling contrast imaging (ECCI) micrographs for Al$_{0.2}$CoNiV alloys annealed at 800, 850, and 900 °C for 1 h. Yellow-dotted lines indicate the boundaries between the recrystallized region and the non-recrystallized region. **j**, **k** Schematics showing the microstructural evolutions from non-recrystallized state to recrystallized state.

array of L2$_1$ particles along the micro-shear bands. The observation of the FCC (γ) and L2$_1$ interface identifies a K–S orientation relationship between them $((111)_{FCC}//(1\bar{1}0)_{BCC}$, $[\bar{1}10]_{FCC}//[111]_{BCC})$ (Supplementary Fig. 6). High-resolution TEM and fast Fourier-transform (FFT) images confirm that the micro-shear bands consist of stacking faults (SFs) and nanotwin (NT) bundles (Fig. 3g, h). Figure 3i shows an atom probe tomography (APT) reconstruction acquired from the non-recrystallized grain, which includes both FCC matrix and an L2$_1$ precipitate. An 1-D concentration profile across these two phases reveals that Al partitions to L2$_1$, while V and Ni partition to FCC, and Co shows no pronounced partitioning. The measured chemical compositions of the FCC matrix and L2$_1$ were 40V–34Co–24Ni–2Al (at%) and 33V–33Co–18Ni–16Al (at%), respectively. Consequently, the precipitates are identified as the (Co,Ni)$_2$VAl-type L2$_1$, so-called Heusler phase.

**Tensile properties**. Figure 4a shows the room-temperature engineering stress–strain curves of our designed MEAs. To emphasize the substantial enhancement in mechanical properties upon the formation of the semicoherent nanoprecipitates, the curve of reference RA sample without the assistance of shear

bands is also presented for comparison. The tensile properties are summarized in Supplementary Table 3. The fully recrystallized A900 alloy exhibits a yield strength of 1050 ± 20 MPa, an ultimate tensile strength of 1480 ± 11 MPa, and a total elongation of 31.7 ± 4.4%, similar to those of the CoNiV alloy with a grain size of ~2 μm[15]. This indicates that the L2$_1$ particles at grain boundaries have little effect on the yield strength. The small decrease in ductility by 6.3% can be attributed to ease of void nucleation and coalescence at interfaces between the rigid L2$_1$ particles and soft FCC matrix[31], because the brittle fracture was not observed (Supplementary Fig. 7). For the A800 alloy, the yield and ultimate tensile strengths are as high as 1500 ± 15 MPa and 1727 ± 22 MPa, respectively. Most notably, we observe a remarkable strength–ductility balance of 1587 MPa × 26.7% for the A850 alloy. These remarkable properties are compared in Fig. 4b, c with other solid-solution or precipitation-strengthened HEAs and MEAs (Supplementary Table 4 for detailed information). The alloys designed in this work exhibit an excellent combination of ultrahigh strength and uniform ductility, surpassing the previously reported HEAs and MEAs having semi-coherent and incoherent phases, and being comparable to the recently reported ultrastrong and ductile coherent L1$_2$ precipitation-strengthened HEAs and MEAs. The mechanical

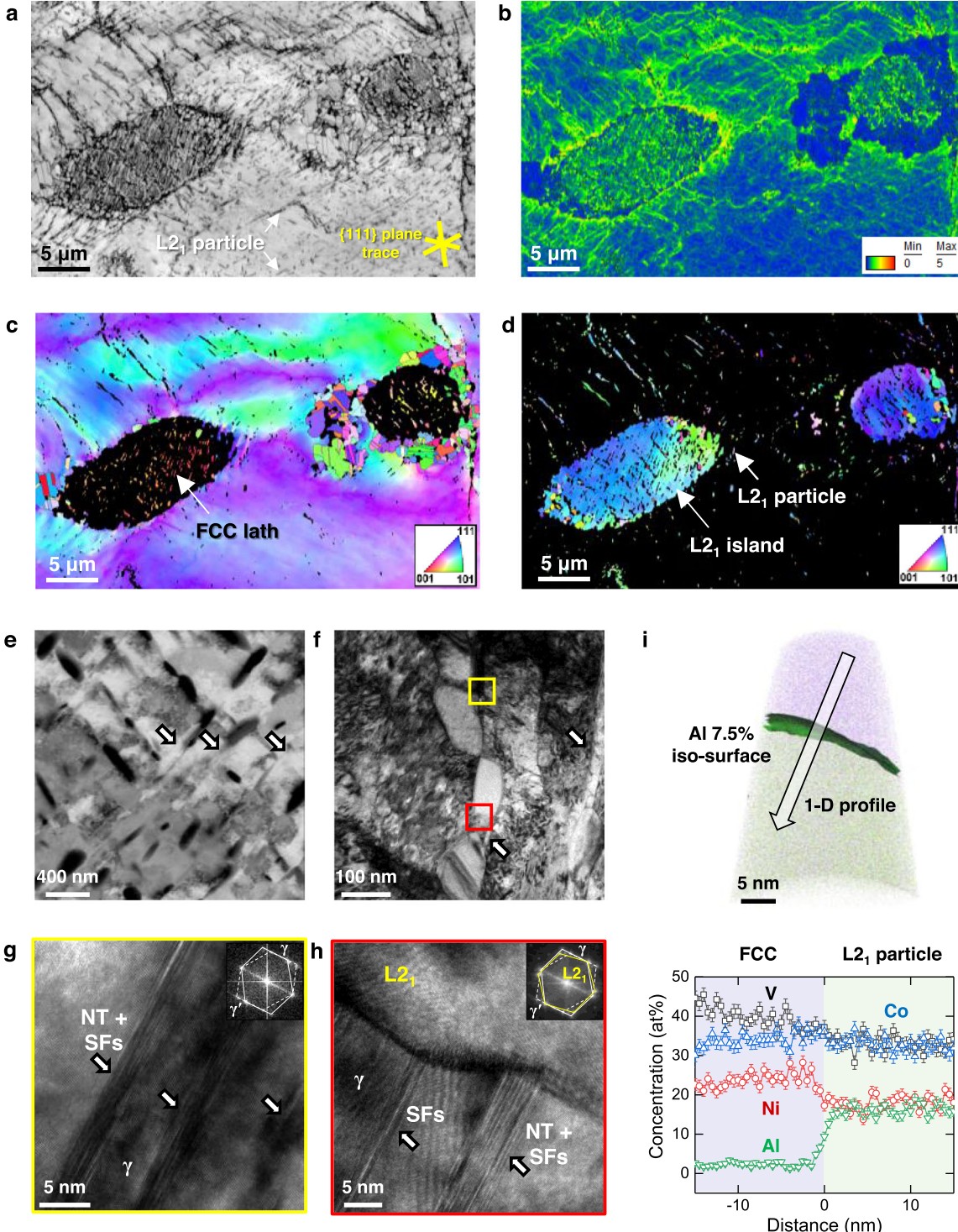

**Fig. 3 The microstructure evolution in the medium-entropy Al$_{0.2}$CoNiV alloy heat-treated at 800 °C for 1 h. a** Electron backscatter diffraction (EBSD) images showing quality (IQ), **b** kernel average misorientation (KAM), inverse pole figure (IPF) maps of **c** face-centered-cubic (FCC) and **d** L2$_1$ phases. **e** Electron-channeling contrast imaging (ECCI) map, **f** transmission electron microscopy (TEM) image of micro-shear bands and L2$_1$ particles in the non-recrystallized FCC grains. **g**, **h** High-resolution TEM and corresponding fast Fourier-transform (FFT) images indicate the micro-shear band consists of stacking faults and nanotwins. **i** Atom probe tomography (APT) tip reconstruction and 1-D profile across the L2$_1$ and FCC matrix. The phase boundary is highlighted by a 7.5 at% Al iso-concentration surface. Each datum point stands for an average concentration measured at a 0.5 nm interval, where error bars indicate a standard deviation.

properties of the present alloys seem to follow a linear trend of the strength–ductility combination of the CoNiV alloys. It should be noted that it is challenging to further enhance the mechanical properties of CoNiV alloys, as the limited process windows in conventional processing restrict the refinement of grains to sizes smaller than ~2 µm[16]. Another strengthening mechanism should be embodied and the present approach enables further improvement in mechanical properties.

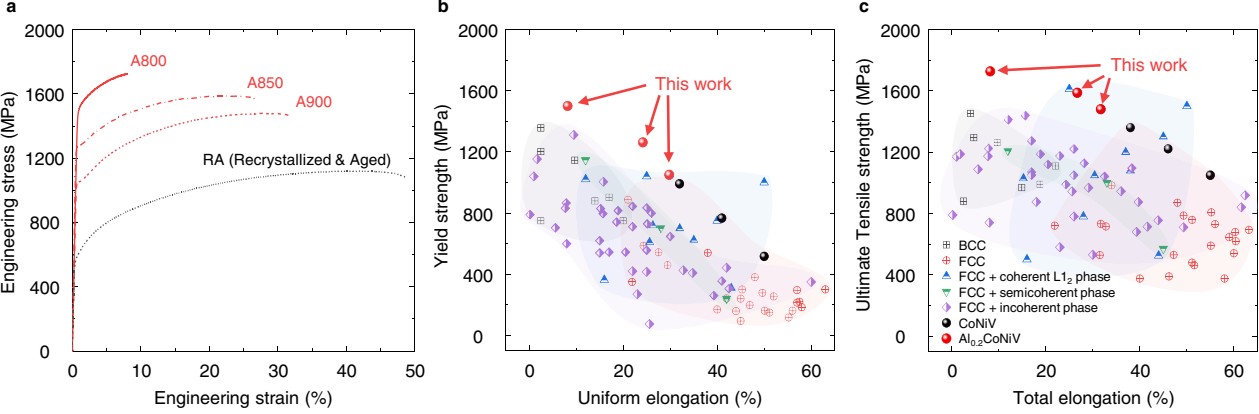

**Fig. 4 Room-temperature mechanical properties of our alloys. a** Engineering tensile stress–strain curves for the annealed $Al_{0.2}CoNiV$ alloys. **b** Overview of yield strength versus uniform elongation, and **c** Ultimate tensile strength versus total elongation values for the current $Al_{0.2}CoNiV$ alloy, compared to single or multiphase high-/medium-entropy alloys. The mechanical properties are summarized in Supplementary Table 3.

## Discussion

Similar to the precipitation of the $L1_2$-ordered FCC phase in the disordered FCC matrix, the $L2_1$ Heusler phase exhibits a high degree of coherency with the disordered BCC matrix[32]. However, in the present $Al_{0.2}CoNiV$ alloy, V promotes the formation of $L2_1$ ordered BCC precipitates in a disordered FCC matrix. The $L2_1$ Heusler phase forms an incoherent interface with the matrix, and thus nucleates at grain boundaries or triple junctions. In addition, intergranular $L2_1$ precipitates grow rapidly as the incoherent interface migrates at a high rate, thereby forming micron-sized coarse particles[18]. In this respect, the coherency between precipitates and the matrix plays a significant role in the formation and stability of uniformly and finely dispersed nanoprecipitates. The formation of homogeneously distributed fine nanoparticles is the key strategy for obtaining ultrastrong ductile precipitation-strengthened alloys.

Even though the FCC and $L2_1$ phases usually form an incoherent interface, they can form a semicoherent interface according to the K–S or Nishiyama–Wassermann (N–W) orientation relationships between FCC and BCC structures. For both relationships, the closest packed planes of each structure are parallel to each other, i.e., $\{111\}_{FCC}//\{110\}_{BCC}$, which correspond to their slip planes. This orientation indicates that $L2_1$ precipitates, nucleated at the slip plane of FCC, prefer to form a semicoherent interface having K–S or N–W relationships. In conjunction with the manipulation of orientation relationships, a strategy to avoid intergranular precipitates is to provide additional nucleation sites by introducing micro-shear bands inside the grains. The cold-rolled and partially recrystallized alloy has a high dislocation density along the micro-shear bands in the non-recrystallized grains. As a wall of a high dislocation density shows higher energy than the lattice, it can act as a heterogeneous nucleation site. Moreover, the elastic field around the dislocation core lowers the activation energy of atomic migration and promotes pipe-diffusion and precipitation at dislocations[33]. Thus, the easy and profound nucleation of the $L2_1$ phase is achieved along $\{111\}$ planes containing deformation-induced defects, which cannot be commonly seen in other types of precipitate-containing alloys.

As stated above, the intergranular $L2_1$ particles of the A900 alloy exhibit an incoherent or only one-sided semicoherent interface with the adjacent disordered FCC grains. In contrast, although not all the interfaces are indexed entirely due to the limited spatial resolution, the intragranular $L2_1$ precipitates in the A800 alloy show a semicoherent interface on all sides having a K–S relationship with the disordered FCC matrix (Supplementary Fig. 5a). In addition, the intragranular $L2_1$ precipitates are well

aligned along the $\{111\}$ plane traces of FCC, corresponding to the slip planes. A semicoherent interface shows a low driving force for migration due to low interface energy and also relatively low mobility compared to incoherent interfaces, thus suppressing the growth of precipitates. In addition to the difference in the type of interfaces, the location of the precipitates contributes to their size difference. As shown in Supplementary Fig. 5b, the precipitates at grain boundaries in the A900 alloy with a recrystallized micro-structure have a larger size than the precipitates inside the grains. The intergranular precipitates have direct access to the grain boundaries that could act as easy diffusion paths assisting the rapid growth of intergranular precipitates. Accordingly, intra-granular $L2_1$ precipitates along the micro-shear bands become finer, with an average size of ~60 nm, as compared to inter-granular precipitates at the grain boundaries, and they are dis-tributed homogeneously inside grains due to micro-shear bands and partial recrystallization. Therefore, micro-shear bands play a key role in the nucleation of nanosized precipitates finely dis-persed inside non-recrystallized grains, while macroshear bands promote the formation of fine recrystallized grains and inter-granular $L2_1$ particles.

The current A800 and A850 alloys exhibit superb mechanical properties with a high-yield strength of 1262 and 1500 MPa, respectively. From the above microstructural observations, the $L2_1$ precipitates significantly contribute to the strengthening, but the amount of this strengthening should be quantified because the high dislocation density also plays a dominant role in the enhancement of strength. For the A800 alloy, the strengthening contributions from solid solution, grain boundary, dislocation, and precipitation strengthening are calculated to be ~383, ~359, ~293, and ~350 MPa, respectively ("Methods" and Supplementary Fig. 8). The calculated total yield strength is 1384 MPa, in reasonable agreement with the experimentally determined yield strength of 1500 MPa. The nanosized intragranular $L2_1$ pre-cipitates combined with high dislocation density lead to remarkable strengthening increments for the A800 and A850 alloys by 25.3% and 11.2%, respectively, compared to the A900 alloy.

Another feature of the designed alloy is that a promising uniform elongation of ~8% is still achieved even at the strongly strengthened state. The semicoherent interfaces were enabled through the K–S orientation relationship, which lowers the interfacial energy and also promotes anisotropic growth along $\{111\}$ planes forming an aspect ratio of 3.03 and 2.83 for the A800 and A850 alloys, respectively[34]. Thus, according to the effective precipitate radius ahead of gliding dislocations, two characteristic

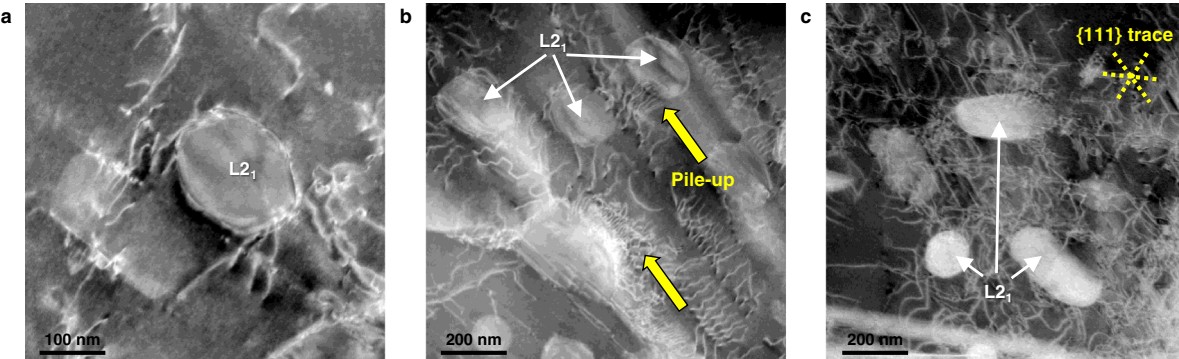

**Fig. 5 High-angle annular dark-field scanning transmission electron microscopy (HAADF-STEM) images showing deformation structures for the Al$_{0.2}$VCoNi alloy annealed at 850 °C for 1 h. a** Orowan bowing mechanism at a L2$_1$ precipitate interface. **b** Dislocations pile up ahead of the L2$_1$ precipitate. **c** Formation of planar-slip dislocation substructures along {111} plane traces and interactions with L2$_1$ precipitates. The deformation structures were observed for specimens deformed to ~1% tensile strain.

mechanisms prevail. The high-angle annular dark-field scanning transmission electron microscopy (HAADF-STEM) images for the deformed A850 alloy reveal that the dislocations approaching the radial direction of cylindrical rod precipitates interact with precipitates by the Orowan bowing mechanism (Fig. 5a). On the other hand, the dislocations pile up at interfaces when they encounter precipitates of large effective radii (Fig. 5b). This pile-up reduces the mean free path of dislocations, leading to significant strain hardening additionally to the Orowan bowing mechanism. For the grain subjected to a relatively large strain (Fig. 5c), the planar dislocation arrays on several independent slip planes construct dislocation network substructures and further reduce the mean free path. Therefore, this unraveled deformation mechanism has a critical role on sustaining the high strain-hardening rate and delaying the necking to high strain and stress levels.

The observation of deformation structures for the specimen deformed to fracture (Supplementary Fig. 9) reveals planar slip with a high dislocation density in the recrystallized grains. This planar-slip character was also found in the equiatomic CoNiV alloy[15], demonstrating that the deformation mechanism of the FCC grains additionally containing ~2 at% Al is similar to that of the CoNiV alloy. However, in non-recrystallized FCC grains, dislocation tangles are also found, although it is challenging to distinguish between the pre-existing dislocations before tensile deformation and the generated ones. Therefore, the A800 sample exhibits low ductility due to the initially high dislocation density and the limited fraction of the dislocation-free recrystallized grains. The fracture surface observations support the conclusion that the L2$_1$ and σ phases do not induce brittle characteristics (Supplementary Fig. 7). Therefore, an effective strategy for achieving a proper balance between strength and ductility is to control the ratio between the recrystallized and non-recrystallized regions equivalent to the ratio between the intergranular and intragranular L2$_1$ precipitates, respectively. The A850 alloy exhibits a remarkable balance of mechanical properties due to the non-recrystallized regions responsible for strength and the recrystallized region responsible for ductility.

In summary, the distinctive ultrahigh strength and ductility are attributed to the microstructure decorated by finely dispersed and semicoherent nanoprecipitates along micro-shear bands. The micro-shear bands, consisting of bundles of NTs and SFs, act as heterogeneous nucleation sites and effectively hinders the formation of coarse and incoherent precipitates at grain boundaries, which have a negligible effect on strengthening. This alloy design and process route enable to modify the size, morphology, and distribution of the precipitates, leading to a remarkable

strength–ductility balance of 1587 MPa and 26.7% surpassing that of previously reported HEAs and MEAs. This work suggests a way to utilize structurally dissimilar precipitates in HEAs and MEAs, where multiprincipal elements coexist and thus can form various types of ordered intermetallic phases.

## Methods

**Ab initio calculations**. The simulations of the phase stability were performed employing DFT calculations using the exact muffin-tin orbitals method[35–40] in combination with the coherent potential approximation[41–43] to account for chemical disorder. The exchange-correlation functional was approximated with the generalized gradient approximation parametrized by Perdew, Burke, and Ernzerhof[44]. The Brillouin zone was sampled with the Monkhorst–Pack method[45,46], where a $k$-points density of 0.125 2π Å$^{-1}$ was employed. The Coulomb screening parameters in the single-site approximation were fixed as $\alpha = 0.75$ and $\beta = 1.14$. The paramagnetic state was modeled with the disordered local moment (DLM) approximation[47,48]. The free energy differences at finite temperature were computed including the configurational entropy in the ideal mixing limit and the electronic entropy utilizing a Fermi smearing.

**Alloy fabrication**. An alloy of Al$_{0.2}$CoNiV nominal composition (at%) and 100 × 35 × 8 mm$^3$ dimensions was cast using vacuum induction melting (model MC100V, Indutherm, Walzbachtal–Wossingen, Germany) and high-purity elements (>99.95%). The blocks were homogenized at 1200 °C for 24 h in evacuated quartz ampules and water quenched. The surfaces of the blocks were pickled in a 20% HCl solution. Subsequently, the samples were cold-rolled until a thickness reduction of 80% was reached, and the resulting 1.5-mm-thick cold-rolled sheets were annealed at 800, 850, and 900 °C in a box furnace for 1 h in an Ar atmosphere, followed by water quenching. To investigate the precipitation behavior excluding the shear bands, the cold-rolled sheets were annealed at 1150 °C for 2 h, followed by water quenching, and subsequently aged at 850 °C for 1 h, followed by water quenching in a box furnace at Ar atmosphere.

**Microstructural characterization**. XRD (Cu K$_\alpha$ radiation, scan rate: 2° min$^{-1}$, scan step size: 0.02°) measurements were performed for crystallographic analyses. The grain size, morphology, and distribution were investigated via EBSD and ECCI, where the specimens were prepared by mechanical polishing using a colloidal silica suspension. The EBSD measurements were performed with a field emission scanning electron microscope (FE-SEM, JEOL, JSM-6500F, USA). The average misorientation of a given point relative to its neighbors was calculated using the KAM approach. The KAM values were calculated up to the third neighbor shell with a maximum misorientation angle of 5° and revealed the deformation-induced local orientation gradients. Additionally, ECCI analyses were carried out using a Zeiss–Merlin instrument (Zeiss Crossbeam 1540 EsB, Zeiss, Oberkochen, Germany). The evolved nanostructures were further investigated via TEM performed on a JEOL JEM-2100F instrument operated at 200 kV. TEM specimens were prepared using the focused ion beam (FIB) lift-out technique in a FEI Helios NanoLab 450 F1 instrument. The chemical composition of each phase was measured via APT (LEAP 4000X HR, Cameca Instruments Inc.) applying the pulsed laser mode at a specimen base temperature of ~50 K. The pulse frequency and energy were 200 kHz and 50 pJ, respectively. The acquired APT data were reconstructed and analyzed using the commercial IVAS® software by Cameca.

**Mechanical tests**. Dog bone-shaped flat specimens were fabricated by electrical discharge machining. The gauge length, width, and thickness of the tensile

specimens were 6.4, 2.5, and 1.5 mm, respectively. Uniaxial tensile tests were carried out at room temperature using a universal testing machine (model: 8801, Instron, Canton, MA, USA) at a crosshead speed of $6.4 \times 10^{-3} \, \text{mm s}^{-1}$. The tensile strains were measured using a digital image correlation (DIC) system with a vision strain gauge system (ARAMIS 5 M, GOM mbH, Germany). Representative data were obtained by averaging three values for each datum point. To reveal the deformation structures in terms of precipitates, tensile specimens were deformed by ~1% of plastic strain and observed by scanning-TEM (STEM, FEI, Talos F200X). TEM specimens were prepared via twin-jet electro-polishing in a solution containing 10 vol.% perchloric acid and 90 vol.% ethanol at a temperature of −20 °C with a potential of 20 V. The samples tensile deformed up to fracture were investigated via ECCI.

**Estimation of strengthening by various mechanisms**. The yield strength of polycrystalline alloys can be obtained as a summation of the four individual contributions: intrinsic friction stress ($\sigma_0$), grain-boundary strengthening ($\Delta\sigma_{GB}$), dislocation strengthening ($\Delta\sigma_\rho$), and precipitation strengthening ($\Delta\sigma_{PH}$), as expressed by Eq. (1). The Al$_{0.2}$CoNiV alloy has a heterogeneous microstructure composed of recrystallized and non-recrystallized regions. To quantify the contribution of each strengthening mechanism taking into account the microstructural heterogeneity, a simple rule of the mixture was used for both regions, as expressed in Eq. (2).

$$\sigma_{0.2} = \sigma_0 + \triangle\sigma_{GB} + \triangle\sigma_\rho + \triangle\sigma_{PH} \tag{1}$$

$$\sigma_{0.2} = f_R\sigma_{R0.2} + (1-f_R)\sigma_{N0.2} \tag{2}$$

where $f_R$ is the fraction of the recrystallized region and $\sigma_R$ and $\sigma_N$ are the yield strengths of the recrystallized and non-recrystallized regions, respectively. The contribution of coarse L1$_2$ islands was not considered because of their low fraction, which did not significantly vary with the annealing temperatures.

In the present Al$_{0.2}$CoNiV alloy, most of the Al added to the CoNiV alloy precipitated out, and only a low concentration of Al (~2 at%) remained dissolved in the disordered FCC solid-solution phase. Hence, the Hall–Petch relationship for single-phase FCC CoNiV was used to calculate the contribution of friction stress and grain-boundary strengthening, as expressed by the following equation[49,50]:

$$\sigma_{0.2} = \sigma_0 + kd^{-1/2} \tag{3}$$

where $k$ and $d$ represent the Hall–Petch coefficient and average grain size, respectively. The equiatomic CoNiV alloy was reported to show severe lattice distortion and consequently high friction stress of 383 MPa, which was attributed to the considerable fluctuation of the bond distance between V and other elements[15]. In addition, according to the interior source model of dislocation generation, the lattice distortion leads to both high friction stress and high Hall–Petch coefficient[51]. Following this model, the CoNiV alloy exhibits a high Hall–Petch coefficient of ~870 MPa μm$^{1/2}$, which implies a high sensitivity of stress to grain size.

The Al$_{0.2}$CoNiV alloy specimens annealed at 800 and 850 °C show partially recrystallized microstructures. A high density of the remaining dislocations in non-recrystallized grains impedes dislocation motion, and thus, significantly contributes to the strengthening of the alloys. The contribution can be quantified using the well-known Taylor hardening equation (Eq. (4))[52]:

$$\triangle\sigma_\rho = M\alpha Gb\rho^{1/2} \tag{4}$$

where $M = 3.06$ is the Taylor factor for the FCC crystal, $\alpha = 0.2$ is a constant for FCC alloys, $G$ is the shear modulus of the CoNiV MEA, and $b$ is the Burger's vector. The average dislocation density ($\rho$) for the non-recrystallized FCC grains, obtained from EBSD KAM analyses, is $1.77 \times 10^{15} \, \text{m}^{-2}$ and $1.41 \times 10^{15} \, \text{m}^{-2}$ for the A800 and A850 samples, respectively. The recrystallized FCC grains show dislocation densities on the order of $10^{12}$–$10^{13} \, \text{m}^{-2}$, making negligible contributions to the strength of about 20–70 MPa.

As stated above, L1$_2$ precipitates at the grain boundaries have little effect on the strengthening. Only the intragranular precipitates can significantly contribute to the improvement of yield strength. The intragranular L1$_2$ precipitates, which have a different crystal structure from the FCC matrix, have a diameter of ~60 nm in the A800 sample, indicating that it is difficult to shear the precipitates by dislocation glide. Thus, these precipitates are expected to contribute to the strengthening through the Orowan bowing mechanisms, as expressed by the following equation[2]:

$$\triangle\sigma_{PH} = M\frac{0.4Gb}{\pi(1-\nu)^{1/2}}\frac{\ln(\frac{2\bar{r}}{b})}{\lambda} \tag{5}$$

where $\nu$ is the Poisson ratio, $\bar{r}$ is the average radius of the precipitates, and $\lambda$ is the average interspacing between precipitates. This equation assumes that all precipitates have an ideal spherical morphology. However, the intragranular L1$_2$ precipitates observed in this work show a rod-like morphology with a specific aspect ratio $h = c/a > 1$, where a and c are the diameter and height of the particle, respectively. Sonderegger et al.[53] modified Eq. (6) for prolate and oblate precipitates by calculating the variation in interparticle spacing according to their

aspect ratio. The following equation expresses how the aspect ratio of the particle affects precipitation strengthening:

$$K = h^{\frac{1}{6}}\left(\frac{2+h^2}{3}\right)^{-\frac{1}{4}} \tag{6}$$

$$\triangle\sigma_{PH} = K^{-1}\triangle\sigma_{PH.spherical} \tag{7}$$

where $K$ is a shape correction factor, $h$ is the aspect ratio of the particle, and $\Delta\sigma_{PH.spherical}$ is the strengthening contribution of spherical particles. The aspect ratios are determined to be 3.03 and 2.83 for the A800 and A850 samples, respectively.

## Data availability
The data that support the findings of this study are available from the corresponding author upon reasonable request.

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

## Acknowledgements

This work was supported by the POSCO Science Fellowship of POSCO TJ Park Foundation; the National Research Foundation of Korea (NRF-2020R1C1C1003554); the Creative Materials Discovery Program of the National Research Foundation of Korea (NRF) funded by the Ministry of Science and ICT (NRF-2016M3D1A1023384); the Korea Institute for Advancement of Technology (KIAT) grant funded by the Korea Government (MOTIE, P0002019, The Competency Development Program for Industry Specialist); Nederlandse Organisatie voor Wetenschappelijk Onderzoek (NWO)/Stichting voor de Technische Wetenschappen (STW), VIDI Grant No. 15707; and German Research Foundation (Deutsche Forschungsgemeinschaft, DFG) under the priority programme 2006 "CCA – HEA".

## Author contributions

T.J.J. and W.S.C. contributed equally to this manuscript. S.S.S. designed the research. D.W.K. and T.J.J. fabricated materials and conducted mechanical tests. W.S.C., G.C., and H.J. carried out TEM and APT experiments. A.F. and F.K. performed ab initio calculations. T.J.J., W.S.C., P.C., and S.S.S. analyzed the data and wrote the manuscript. All authors reviewed and contributed to the final manuscript.

## Competing interests

The authors declare no competing interests.
