## [Peer Review File · Nature Communications]

REVIEWER COMMENTS

Reviewer #1 (Remarks to the Author):

This manuscript describes the development of an ultrastrong MEA strengthened mainly by L21 precipitates. The properties are very good, high strength and good ductility were achieved simultaneously, which is rarely seen. In addition, such properties were not achieved by ordered coherent precipitates such as L12, which has been known to have such potential. Thus, this work may demonstrate a different path to great HEA/MEAs. The experimental works were done nicely, too. Particularly the EBSD and ECCI works. However, some important issues need to be addressed. Most importantly, the reason for the simultaneous high strength and good ductility has not been clearly discussed and revealed.

1. The properties and microstructures of reference state should be provided at least in the supplement so that the readers have an idea of the contributions of the various factors. This includes (1) the homogenized state (is it really single phase? how strong is it? this reflects the solid solution strengthening). (2) the as-rolled state (how strong is it? this tells solid solution strengthening plus work hardening) (3) details of the “HA” state (author did not specify its treatment condition in Fig 4, also is its microstructure the one shown in Fig S1?)

2. Although not much demonstrated in HEA/MEAs, the use of non-coherent or semi-coherent precipitates to achieve good properties is not new, and has been demonstrated in steel and Al alloys. An example is given below.

<https://www.nature.com/articles/nature14144>

Important works of this kind should not be omitted and must be properly cited.

3. The VCoNi alloy has been demonstrated to have great properties if the grains were refined to 2 μm . Your 900C annealed alloy also has similar grain size and properties. So it seems the large amount of precipitates barely have any effect on the properties, both in terms of strength and ductility. This is not reasonable and should be explained.

4. Supplementary Fig. 4 should be further labeled and explained. The color labels in Fig. 4a are not related to the Fig at all. What do the green area (L21?) and the red line represent? The authors used this Fig to evidence semi-coherent interface. Is there any direct evidence for the semi-coherent interface? (e.g. HRTEM)

5. The results from some important publications are apparently missing in Fig. 4b and 4c. For example, CTLiu's two science papers. It is suggested that the authors do a more thorough check on important papers.

6. The ductility of the alloys was not explained clearly. For alloys with such high strength, an exceptionally high work hardening capability is needed to postpone necking to such high strain. This is

clearly an unique property of your alloys and should be discussed. What sustains the high work hardening rate in your alloys? What is the design strategy to achieve it? The authors need to unravel these so that the manuscript will justify the broad impact needed for this top flag journal.

7. L93-97: the effect of pseudogap and its position relative to E_f on the stability of L21 phase should be further explained for clarity.

Reviewer #2 (Remarks to the Author):

The authors designed a new alloy by adding Al to the VCoNi medium-entropy alloy. While adding Al to MEAs and HEAs is commonly used to form second phases with coherent/incoherent interfaces, adding Al to VCoNi led to the precipitation of a semi-coherent phase, making it a very interesting study. The authors successfully used pre-deformation and subsequent annealing to control the size, density, and distribution of the semi-coherent precipitates. They did a systematic study on the microstructure characterization and mechanical properties of the designed alloy.

However, there are some main issues regarding (1) the underlying mechanism of precipitation, so-called "shear-band driven precipitate dispersion", and (2) the assessment of mechanical properties that need to be addressed in order to improve the manuscript. All comments and questions, including the issues mentioned above, are summarized as follows.

Lines 98-99: homogenization and aging conditions ("1200 °C for 24h and 900 °C for 1h, respectively") do not match the condition mentioned in Supp. Data, Fig.1 (850 °C for 1h)!

I assume this sample is the "HA" sample of which mechanical properties were compared with those of A800, A850, A900 samples later in the manuscript. If so, it is better to introduce "HA" here, not in the later part.

Lines 104-107: "as shown in Fig. 1c, here we introduced a plenty of nucleation sites ...", this is a bit strange expression because Fig.1c is a simple thermomechanical processing schedule, and it does not show any microstructure-related information!

Fig. 2 and the term "shear band": authors described that after annealing, (macroscopic) shear bands were replaced by fine recrystallized grains (Fig. 2a), which is a reasonable assessment. Then they claimed that semi-coherent precipitates in non-recrystallized regions also nucleated along shear bands. The latter assessment was widely used throughout the manuscript as the underlying mechanism of semicoherent phase formation. But what is the definition of shear bands in the non-recrystallized regions? Are authors referring to a kind of micro-shear bands? If so, I could not find any evidence or clear indication of micro-shear bands in Fig. 2 (or Fig. 3 in the later part). As far as I know, shear bands are not necessarily aligned along specific crystallographic planes. Also, nanotwin and stacking fault bundles (as the nucleation sites for L21 : Fig.3 g-h) can be generated by the deformation without the contribution of shear banding.

Regarding Fig.2, I also recommend providing the deformation microstructure after 80%CR (before annealing). Giving details of the deformation microstructure makes it easier to understand what happens during the subsequent annealing heat-treatment.

Line 138: “will be further discussed below”, this sentence may indicate that (K-S) orientation relationship is discussed immediately after this paragraph, but in fact it is discussed in a later part.

Lines: 150-151: “L21 precipitates start to form at the grain boundaries or triple junctions ...” a fraction of L21 precipitates in A900 sample also formed inside the recrystallized grains as isolated precipitates or connected to annealing twin boundaries. This seems to be a characteristic of L21 phase and should not go unnoticed in this part.

Lines 151-152: “L21 particles exhibit an incoherent or only one-sided semicoherent interface ... (Supplementary Fig. 4)”, assuming that an interface between L21 and FCC matrix follows (K-S) orientation relationship, does it necessarily mean that the interface is semicoherent?

Lines 164-165: “... evident green lines along {111} plane trace ... along the shear bands.”, why those green-line features in KAM map that appear along different {111} plane traces are considered as shear bands?

Lines 188-193: “A proximity histogram across these two phases ... so-called Heusler phase.” I think using the proximity histogram is suitable when particles (or clusters) are fully embedded inside the APT tip. In Fig. 3i, it is better to provide a 1-D composition profile across a cylindrical ROI perpendicular to the interface and having at least 20nm length in each phase. That may be a better estimation for the elemental distribution in each phase.

In addition, it seems that the measured composition of L21 (32V-32Co-18Ni-18Al) deviated from the stoichiometry of so-called Heusler phase (Co,Ni)₂VAI. Is there any specific reason for that?

Fig. 4 and the assessment of mechanical properties: it is clear that VCoNiAl_{0.2} shows an excellent combination of strength-ductility, but overall it does not seem that the alloy containing semicoherent precipitates can overcome the strength-ductility trade-off. In particular, the ductility of A800 sample having the highest fraction of semicoherent precipitates dropped substantially compared to A850 and A900. In this regard, semicoherent precipitates seem to have a similar degrading effect on ductility as a TCP phase does (e.g. σ phase).

In Fig. 4b-c, the author made a nice effort to compare the mechanical properties of VCoNiAl_{0.2} with other references. As an additional reference, I recommend adding the data associated with the single-phase VCoNi system that was previously studied by the authors (Ref. 14). Here, the critical question is that whether the strength-ductility combination in the presence of semicoherent precipitates (VCoNiAl_{0.2}) surpasses that of single-phase VCoNi alloy or not.

Lines 239-245: it may not be a good idea to generally discuss a list of defects (dislocation, dislocation walls, NTs, SFs) that can possibly be a nucleation site for precipitation.

Aside from that, all those defects mentioned above can form along {111} planes without the presence/contribution of shear bands. Is it really necessary to relate the underlying precipitation mechanism to the so-called shear-band driven precipitation?

Lines 246-247: "These underlying mechanisms ... are confirmed in Supplementary Fig. 4", This is a generalized statement. I don't think Fig. 4 alone reveals any underlying mechanisms.

Lines 253-258: as the authors discussed, the smaller size of intragranular L21 in A800 can be related to their semicoherent interfaces (low mobility). However, I think another important factor is the location of precipitation in A800. The fine intragranular precipitates in A800 formed inside the non-recrystallized grains (far from GBs). In contrast, in A900 with recrystallized microstructure, intergranular precipitates had direct access to the GBs that could act as easy diffusion paths assisting the rapid growth of intergranular precipitates. For comparison, there were some in-grain precipitates in A900 (Supp. Fig. 4b), which were clearly smaller than the intergranular precipitates, indicating that access to the GB is crucial for the precipitate growth.

Lines 261-272: the authors put efforts into estimating the contribution of each strengthening mechanism separately, although the methods and calculations they used and described in Supp. Data seem to be oversimplified. But assuming that such estimation and the corresponding Fig. 5 (Supp. Data) are accurate enough, this means that in A850 having the best combination of strength-ductility, only ~10% of strength originates from the semicoherent precipitates in non-recrystallized regions. If so, the strategy of using semicoherent-precipitation strengthening becomes a minor factor, and authors are not able to claim that the excellent properties of A850 are due to their proposed strategy.

Line 284-285: "the A800 sample exhibits low ductility due to the initially high dislocation density and the limited fraction of the dislocation-free recrystallized grains", this is a reasonable statement, but the degrading effect of precipitates on ductility cannot be simply ignored. Have authors conducted any fracture surface analysis? I understand carrying out experiments during the COVID-19 pandemic can be difficult, but it would be interesting to see whether the fracture mechanism in this sample is related to the semicoherent particles or not.

Other comments:

- In the introduction part, the authors described the advantage of using coherent precipitates over incoherent precipitates (or TCP phases). Then they designed an alloy having semicoherent precipitates and examined the microstructures and mechanical properties. But after looking at the results and discussion, it is not yet convincing whether there is a big advantage of using semicoherent phases over incoherent phases.

However, I do think that the semicoherent L21 in VCoNiAl_{0.2} has a unique behavior, i.e. the easy/profound nucleation of L21 along {111} planes containing deformation-induced defects, which cannot be commonly seen in other types of precipitate-containing alloys. I think the authors can

elaborate on this fact to improve the discussion part.

- Phase identification of VCoNiAl_{0.2} alloy via XRD analysis is quite important (evidence for L21 phase). I recommend showing XRD profile as a main figure in the manuscript, not as supplementary data.

- Some detailed information needs to be added to some figure captions: for example, red lines in Fig. 4 (Supp. Data) indicates semicoherent interfaces or “HA” in Fig. 4 stands for “homogenized & aged”, or symbols σ_{PH} , etc. in Fig. 5 (Supp. Data) represent ... It is much easier to understand a figure as it stands without searching for necessary information in the text.

Reviewer #3 (Remarks to the Author):

The paper “Shear band-driven precipitate dispersion for ultrastrong ductile medium-entropy alloys” presents characterization of a complex concentrated alloy Al_{0.2}CoNiV. Tensile deformation testing of the alloy revealed very high yield strength of 1260 MPa in combination with sufficient ductility of 27%. This is definitely an interesting finding. High strength of the alloy is attributed to the precipitation strengthening by finely dispersed particles of L21 Heusler phase. These particles have well defined orientation relationship with the disordered fcc matrix and form semicoherent interfaces with the matrix. Although the results reported in the paper, in particular results of the microstructure investigations, are interesting there are some points which have to be improved and/or explained better.

1. According to the convention used for designating of complex concentrated alloys the constituting elements should be mentioned in the alphabetic order. For the present alloy it reads Al_{0.2}CoNiV.

2. It is surprising that the authors do not present results of fractography, i.e. surfaces of samples broken in the tensile deformation test, in order to analyze the mode of deformation of Al_{0.2}CoNiV alloy and to examine whether it exhibits transgranular or intergranular fracture.

3. Recrystallized grains contain a lot of annealing twins visible in Fig. 2i and shear bands with L21 particles contain nanotwins. However it seems that contribution of twins is not included in the strengthening model presented in lines 348-414.

4. It is mentioned in the manuscript (lines 153-154) that the L21 particles occur at grain-boundaries or triple junctions in the recrystallized fcc region. But in Fig. 2i the L21 particles can be seen not only in grain boundaries but also inside grains.

5. Fig. 4: What is the difference between uniform elongation and total elongation?

6..In the text of the manuscript and also in the Supplementary Table 3 the authors use the term “tensile

strength” but actually they probably mean “ultimate tensile strength”. This should be corrected everywhere in the paper.

7. Experimental uncertainties should be mentioned in the relative concentrations of phases in Supplementary Table I.

8. All uncertainties in Supplementary Table 2 and Table 3 have to be rounded to one standard digit and measured data have to be rounded to the order of the last significant digit of the uncertainty.

Reviewer #1 (Remarks to the Author):

This manuscript describes the development of an ultrastrong MEA strengthened mainly by $L2_1$ precipitates. The properties are very good, high strength and good ductility were achieved simultaneously, which is rarely seen. In addition, such properties were not achieved by ordered coherent precipitates such as $L1_2$, which has been known to have such potential. Thus, this work may demonstrate a different path to great HEA/MEAs. The experimental works were done nicely, too. Particularly the EBSD and ECCI works. However, some important issues need to be addressed. Most importantly, the reason for the simultaneous high strength and good ductility has not been clearly discussed and revealed.

Question #1> The properties and microstructures of reference state should be provided at least in the supplement so that the readers have an idea of the contributions of the various factors. This includes (1) the homogenized state (is it really single phase? how strong is it? this reflects the solid solution strengthening). (2) the as-rolled state (how strong is it? this tells solid solution strengthening plus work hardening) (3) details of the “HA” state (author did not specify its treatment condition in Fig 4, also is its microstructure the one shown in Fig S1?)

Reply #1> We appreciate the helpful comment on the reference states of the $Al_{0.2}CoNiV$ alloy. We have conducted additional experiments to show the properties and microstructures of the reference states. The results are embodied in Supplementary Fig. 1 and 2. Here are the responses to each state the reviewer suggested:

(1) The homogenized state: The homogenized state, heat-treated at 1200 °C for 24 h, of the current alloy consists of disordered FCC matrix and Al-rich $L2_1$ island. The average grain size of FCC matrix was measured as ~250 μm . The yield strength, tensile strength, and ductility of the homogenized state sample are 495 MPa, 1012 MPa, and 41.9 %, respectively. The yield strength is higher by ~60 MPa than the calculated value from the Hall-Petch relationship of the

CoNiV alloy as expressed in Eq. 3. This difference might be attributed to the Al-rich L2₁ islands.

The stress-strain curves and microstructures of the homogenized states have been added in

Supplementary Fig. 1.

- **Supplementary Fig. 1 a** Tensile stress-strain curve of the homogenized state, **b, c** SEM image and corresponding EDS elemental maps exhibiting L2₁ islands

(2) The as-rolled state: The cold-rolled state shows yield strength, ultimate tensile strength, and ductility of 1741 MPa, 1959 MPa, and 9.4 %, respectively. The microstructure of the as-rolled state shows thick macro-shear bands distributed along ~10–45 degrees to the rolling direction. These macro-shear bands play a significant role in producing fine recrystallized FCC grains during subsequent annealing processes. Inside FCC grains alongside macro-shear bands, there are micro-shear bands which consist of mostly dislocation bands, which will be described in Question #9 in detail. The stress-strain curves and microstructures of the cold-rolled states have been added in Supplementary Fig. 2.

- **Supplementary Fig. 2 a** Tensile stress-strain curve of the cold-rolled state, **b, c** optical and SEM images showing macro-shear bands along ~10–45 degrees to the rolling direction, **d-f** EBSD IPF, IQ, and KAM maps showing macro-shear bands and micro-shear bands indicated by yellow and white arrows, respectively.

(3) The “HA” state: Thank you very much for pointing out the missing information. The “HA” state represents a specimen annealed at 1150 °C for 2 h (homogenization) after cold-rolling and subsequently aged at 850 °C for 1 h. It seems to be confusing to readers because the term “homogenization” is used ambiguously in two different processes, *i.e.*, homogenization of the as-cast alloy (1200 °C for 24 h) and homogenization of the cold-rolled alloy (1150 °C for 2 h). Therefore, to specify each process, we have changed the term “HA” to “RA” which means Recrystallization and Aging. Accordingly, we have modified Figure 1c, Supplementary Fig. 3, and relevant statement on **pages 6 and 18** in the manuscript as follows.

- **Fig. 1 Material and process design.** ... c Schematics of the thermomechanical processing for precipitation strengthened $\text{Al}_{0.2}\text{CoNiV}$ alloys.
- After subsequent recrystallization and aging process (1150 °C for 2 h and 850 °C for 1 h, respectively), referred to as RA, ...
- **Alloy fabrication.** ... To investigate the precipitation behavior excluding the shear bands, the cold-rolled sheets were annealed at 1150 °C for 2 h, followed by water quenching and subsequently aged at 850 °C for 1h, followed by water quenching in a box furnace at Ar atmosphere.

- **Supplementary Fig. 3** a, b Low- and high- magnification SEM images, and c EDS elemental maps exhibiting L2₁ island of the RA sample.

Question #2> Although not much demonstrated in HEA/MEAs, the use of non-coherent or semi-coherent precipitates to achieve good properties is not new, and has been demonstrated in steel and Al alloys. An example is given below. Important works of this kind should not be omitted and must be properly cited. <https://www.nature.com/articles/nature14144>.

Reply #2> We appreciate for introducing the research using dissimilar precipitates and shear bands to make ultrastrong low-density steels with large ductility. We agree with the suggestion that studies on deformation-induced precipitation and property improvement should be cited properly. Therefore, we have added the following references and relevant descriptions regarding the low-density steel and Al alloy on **page 3** in the introduction part.

- ... Nevertheless, to conquer the severe brittleness of dissimilar phases with a matrix, one feasible way is to manipulate semicoherent precipitates by avoiding the formation of incoherent interfaces with the matrix. The lower interfacial energy of semicoherent precipitates than incoherent ones reduces their coarsening, while the homogeneous distribution in nanoscale can be embodied by introducing additional nucleation sites in lattices, as reported in low-density steel and Al alloys^{11,12} ...
- **Reference 11**. Kim, H. & Kim, N. J. Brittle intermetallic compound makes ultrastrong low-density steel with large ductility. *Nature* **518**, 77-79 (2015).
- **Reference 12**. Cai, Y., Lang, Y., Cao, L. & Zhang, J. Enhanced grain refinement in AA7050 Al alloy by deformation-induced precipitation. *Mater. Sci. Eng. A* **549**, 100-104 (2012).

Question #3> The VCoNi alloy has been demonstrated to have great properties if the grains were refined to 2 um. Your 900C annealed alloy also have similar grain size and properties. So it seems the large amount of precipitates barely have any effect on the properties, both in terms of strength and ductility. This is not reasonable and should be explained.

Reply #3> We appreciate the comment, as it gives us the opportunity to explain the relationship between tensile properties and microstructures of the A900 alloy compared to the CoNiV alloy. As mentioned in material and process design part, precipitates at grain boundaries do not significantly contribute to strengthening because it is difficult for dislocations to directly interact with precipitates during dislocation gliding. Instead, dislocations rather readily pile up in front of precipitates at the pre-existing pile-up site, *i.e.*, grain boundary. Therefore, the A900 alloy has a similar yield strength of ~1050 MPa with ~1000 MPa for the CoNiV alloy as the reviewer exactly pointed out. To utilize this precipitate as a strengthening factor, thus, we introduced the intragranular L2₁ precipitates instead of intergranular ones such in A800 and A900 alloys. This is the main objective of our study.

In terms of ductility, the total elongation of the A900 alloy is lower by 6.3% compared to the CoNiV alloy having a similar grain size. We have conducted additional experiments to investigate fracture characteristics of the A900 alloy. The ductile-dimpled fracture was observed without any brittle appearance such as intergranular fracture. In other words, the little decrease of ductility might be attributed to the ease of void nucleation and coalescence at FCC/L2₁ interfaces. For the A900 alloy, the L2₁ at GB has an incoherent interface with FCC grains and a semicoherent interface as well. Therefore, the following sentences and reference have been added to support the explanation of tensile properties on page 11, and the following fractographic images have been added in Supplementary Fig. 7.

- ... The fully recrystallized A900 alloy exhibits a yield strength of 1050 ± 20 MPa, an ultimate tensile strength of 1480 ± 11 MPa, and a total elongation of $31.7 \pm 4.4\%$, similar to those of the CoNiV alloy with a grain size of $\sim 2 \mu\text{m}^{15}$. This indicates that the L2₁ particles at grain boundaries have little effect on the yield strength. The small decrease in ductility by 6.3% can be attributed to an ease of void nucleation and

coalescence at interfaces between the rigid L₂₁ particles and soft FCC matrix³⁰, because brittle fracture was not observed (Supplementary Fig. 7). ...

- **Supplementary Fig. 7** Tensile fractographies of the **a** A800, **b** A850, and **c** A900 alloys. Only ductile-dimple fracture was observed
- **Reference 30.** Stone, R. H. V., Cox, T. B., Low, J. R. & Psioda, J. A. Microstructural aspects of fracture by dimpled rupture. *International Metals Reviews* **30**, 157-180 (1985).

Question #4> Supplementary Fig. 4 should be further labeled and explained. The color labels in Fig. 4a is not related to the Fig at all. What do the green area (L₂₁?) and the red line represent? The authors used this Fig to evidence semi-coherent interface. Is there any direct evidence for the semi-coherent interface? (e.g. HRTEM)

Reply #4> We appreciate your helpful comments. White-gray, turquoise (green), and yellow color labels represent FCC, L₂₁, and σ phase, respectively. Some of the small L₂₁ particles in FCC matrix were not indexed properly due to the limited spatial resolution in EBSD analysis. The red line represents the interfaces satisfying K-S relationship between FCC and L₂₁. Similar to the particle indexing, some interfaces were not clearly identified for the A800 alloy. The indication of the red line had been omitted, thus we have added it in the upper-left corner of

Supplementary Fig. 5a. The direct evidence for the semi-coherent interface can be found in the HRTEM image of Fig. 3h in the original submission. To identify more accurately, we have added the magnified HRTEM image and corresponding inverse FFT image in Supplementary Fig. 6.

- ... The observation of the FCC (γ) and L2₁ interface identifies a K-S orientation relationship of them ($(111)_{\text{FCC}} // (1\bar{1}0)_{\text{BCC}}, \bar{1}10]_{\text{FCC}} // [111]_{\text{BCC}}$) (Supplementary Fig. 6) ...

- **Supplementary Fig. 5** EBSD IQ maps of annealed $\text{Al}_{0.2}\text{CoNiV}$ alloys superimposed by phase color. White-gray, turquoise, and yellow colors represent FCC, L2₁, and σ phases, respectively. **a** non-recrystallized region of the alloy annealed at 800 °C and **b** recrystallized region of the alloy annealed at 900 °C. Red line indicates FCC and L2₁ phases having a K-S orientation relationship.

- **Supplementary Fig. 6 a** High-resolution TEM and corresponding FFT images, **b** inverse FFT image showing semi-coherent interface between γ and L2₁.

Question #5> The results from some important publications are apparently missing in Fig. 4b and 4c. For example, CT Liu's two science papers. It is suggested that the authors do a more thorough check on important papers.

Reply #5> We appreciate for helpful comment and suggestions regarding precipitation-hardened FCC-HEAs/MEAs with excellent mechanical properties using coherent L_{12} precipitates and various strategies, such as MBIP (micro-band induced plasticity)^{S19} and nanoscale disordered interfaces^{S20}. In addition to the reviewer's suggestions, we have added the properties of CoNiV alloy having a single FCC phase with no precipitates¹⁵. This comparison enables the discussion of the role of L_{21} precipitates in FCC matrix for CoNiV alloy. Only a few studies reported on HEAs/MEAs with excellent mechanical properties having an incoherent or semicoherent interface with a disordered FCC matrix. We think that one of the novelties of this study is to show the possibility of ultrastrong and ductile HEAs/MEAs by precipitation of a dissimilar phase. To emphasize this novelty, we have modified the grouping "FCC+FCC/ L_{12} ", "FCC/ L_{12} +BCC/B2", and "FCC+Complex phase" into "FCC+coherent L_{12} phase", "FCC+semicoherent phase", and "FCC+ incoherent phase". The following sentences and figures have been revised on page 12 in the manuscript.

- ... These remarkable properties are compared in Fig. 4b, c with other solid-solution or precipitation-strengthened HEAs and MEAs (Supplementary Table 4 for detailed information). The alloys designed in this work exhibit an excellent combination of ultrahigh strength and uniform ductility, surpassing the previously reported HEAs and MEAs having semicoherent and incoherent phases, and being comparable to the recently reported ultrastrong and ductile coherent \$L_{12}\$ precipitation-strengthened HEAs and MEAs. ...

- **Fig. 4 Room-temperature mechanical properties of our alloys.** ... b Overview of yield strength versus uniform elongation, and c **Ultimate** tensile strength versus total elongation values for the current **Al_{0.2}CoNiV** alloy, compared to single or multiphase high-/medium-entropy alloys. ...
- **Supplementary References 19.** Yang, T. *et al.* Multicomponent intermetallic nanoparticles and superb mechanical behaviors of complex alloys. *Science* **362**, 933 (2018).
- **Supplementary References 20.** Yang, T. *et al.* Ultrahigh-strength and ductile superlattice alloys with nanoscale disordered interfaces. *Science* **369**, 427 (2020).
- **Supplementary References 37.** Sohn, S. S. *et al.* Ultrastrong Medium-Entropy Single-Phase Alloys Designed via Severe Lattice Distortion. *Adv. Mater.* **31**, e1807142 (2019).

Question #6> The ductility of the alloys were not explained clearly. For alloys with such high strength, an exceptionally high work hardening capability is needed to postpone necking to such high strain. This is clearly a unique property of your alloys and should be discussed. What sustains the high work hardening rate in your alloys? What is the design strategy to

achieve it? The authors need to unravel these so that the manuscript will justify the broad impact needed for this top flag journal.

Reply #6> We sincerely appreciate the reviewer for providing constructive suggestions that should be discussed to unravel the role of precipitates on mechanical responses. To address the suggestion, we have conducted additional tensile tests to ~1% plastic strain and the deformation structures were analyzed by STEM observation for the A850 alloy. The STEM images have been added in Fig. 5a-c as follows.

- **Fig. 5 HAADF-STEM images showing deformation structures for the $\text{Al}_{0.2}\text{VCoNi}$ alloy annealed at 850 °C for 1 h. a** Orowan bowing mechanism at a L_{21} precipitate interface. **b** Dislocations pile-up ahead of the L_{21} precipitate. **c** Formation of planar-slip dislocation substructures along $\{111\}$ plane traces and interactions with L_{21} precipitates. The deformation structures were observed for specimens deformed to ~1% tensile strain.

The semicoherent interfaces were enabled through the K-S orientation relationship, which lowers the interfacial energy and also promotes anisotropic growth along $\{111\}$ planes forming an aspect ratio of 3.03 and 2.83 for the A800 and A850 alloys, respectively³⁴. Thus, according to the effective precipitate radius ahead of gliding dislocations, two characteristic mechanisms prevail. The high-angle annular dark-field scanning transmission electron microscopy

(HAADF-STEM) images for the deformed A850 alloy reveal that the dislocations approaching the radial direction of cylindrical rod precipitates interact with precipitates by the Orowan bowing mechanism (Fig. 5a). On the other hand, the dislocations pile-up at interfaces when they encounter precipitates of large effective radii (Fig. 5b). This pile-up reduces the mean free path of dislocations, leading to significant strain hardening additionally to the Orowan bowing mechanism. For the grain subjected to a relatively large strain (Fig. 5c), the planar dislocation arrays on several independent slip planes construct dislocation network substructures and further reduce the mean free path. Therefore, this unraveled deformation mechanism has a critical role on sustaining the high strain-hardening rate and delaying the necking to high strain and stress levels. This paragraph has been added on page 16 in the revised manuscript, and the following reference has been added in the manuscript.

- **Reference 34.** Luo, C. P. & Weatherly, G. C. The invariant line and precipitation in a Ni-45 wt% Cr alloy. *Acta Metall.* **35**, 1963-1972 (1987).

Question #7> L93-97: the effect of pseudogap and its position relative to E_f on the stability of L2₁ phase should be further explained for clarity.

Reply #7> We thank the reviewer for this comment, which gives us the opportunity to elaborate more on the electronic structure origin of the stabilization of the L2₁ phase. In general, pseudogaps in the density of states (*i.e.*, regions where the density is almost zero) whose position is close to the Fermi level indicate very stable compounds, because more electronic states can be accommodated at lower energies. We impute the higher stability of the L2₁ phase over the other ordered compounds to the opening of such pseudogaps, as shown in Fig. 1b of the manuscript. We also observe that the pseudogap moves closer to the Fermi level with increasing Co content on the (Co,Ni) sublattice of the precipitate, further increasing its stability.

This relates to the experimental observation of Co-rich L2₁ precipitates from the atom-probe tomography experiments. Accordingly, the following sentences have been added to the manuscript on page 5.

- ... The stabilization of this phase originates from the opening of a pseudogap in the electronic density of states of the L2₁ phase (Fig. 1b) in proximity to the Fermi level (E_F), which is a common indication of a compound with high formation energy because more electronic states can be accommodated at lower energies. As for the stoichiometry of this L2₁ phase, we note that the larger the Co concentration, the closer the pseudogap to E_F , signaling increased stability of the L2₁ phase with increasing amount of Co in this precipitate (See Supplementary Note for detailed information).

Reviewer #2 (Remarks to the Author):

The authors designed a new alloy by adding Al to the VCoNi medium-entropy alloy. While adding Al to MEAs and HEAs is commonly used to form second phases with coherent/incoherent interfaces, adding Al to VCoNi led to the precipitation of a semi-coherent phase, making it a very interesting study. The authors successfully used pre-deformation and subsequent annealing to control the size, density, and distribution of the semi-coherent precipitates. They did a systematic study on the microstructure characterization and mechanical properties of the designed alloy.

However, there are some main issues regarding (1) the underlying mechanism of precipitation, so-called "shear-band driven precipitate dispersion", and (2) the assessment of mechanical properties that need to be addressed in order to improve the manuscript. All comments and questions, including the issues mentioned above, are summarized as follows.

Question #8> Lines 98-99: homogenization and aging conditions ("1200 °C for 24h and 900 °C for 1h, respectively") do not match the condition mentioned in Supp. Data, Fig.1 (850 °C for 1h)! I assume this sample is the "HA" sample of which mechanical properties were compared with those of A800, A850, A900 samples later in the manuscript. If so, it is better to introduce "HA" here, not in the later part.

Reply #8> Thank you very much for pointing out the wrong information. The response to this suggestion is described in Reply #1. To sum up, the "HA" state represents a specimen annealed at 1150 °C for 2 h (homogenization) after cold-rolling and subsequently aged at 850 °C for 1 h. We have corrected the condition in the manuscript. In addition, it seems to be confusing to readers because the term "homogenization" is used ambiguously in two different processes, *i.e.*, homogenization of the as-cast alloy (1200 °C for 24 h) and homogenization of the cold-

rolled alloy (1150 °C for 2 h). Therefore, to specify each process, we have changed the term “HA” to “RA” which means Recrystallization and Aging. Accordingly, we have modified Figure 1c and relevant statement pages 6 and 18 in the manuscript as follows.

- **Fig. 1 Material and process design.** ... c Schematics of the thermomechanical processing for precipitation strengthened $\text{Al}_{0.2}\text{CoNiV}$ alloys.
- ... After subsequent recrystallization and aging process (1150 °C for 2 h and 850 °C for 1 h, respectively), referred to as RA, ...
- **Alloy fabrication.** ... To investigate the precipitation behavior excluding the shear bands, the cold-rolled sheets were annealed at 1150 °C for 2 h, followed by water quenching and subsequently aged at 850 °C for 1h, followed by water quenching in a box furnace at Ar atmosphere.

- **Supplementary Fig. 3 a, b** Low- and high- magnification SEM images, and **c** EDS elemental maps exhibiting L2₁ island of the **RA** sample.

Question #9> Lines 104-107: “as shown in Fig. 1c, here we introduced a plenty of nucleation sites ...”, this is a bit strange expression because Fig.1c is a simple thermomechanical processing schedule, and it does not show any microstructure-related information! Fig. 2 and the term “shear band”: authors described that after annealing, (macroscopic) shear bands were replaced by fine recrystallized grains (Fig. 2a), which is a reasonable assessment. Then they claimed that semi-coherent precipitates in non-recrystallized regions also nucleated along shear bands. The latter assessment was widely used throughout the manuscript as the underlying mechanism of semicoherent phase formation. But what is the definition of shear bands in the non-recrystallized regions? Are authors referring to a kind of micro-shear bands? If so, I could not find any evidence or clear indication of micro-shear bands in Fig. 2 (or Fig. 3 in the later part). As far as I know, shear bands are not necessarily aligned along specific crystallographic planes. Also, nanotwin and stacking fault bundles (as the nucleation sites for L2₁: Fig. 3g-h) can be generated by the deformation without the contribution of shear banding. Regarding Fig. 2, I also recommend providing the deformation microstructure after 80%CR (before annealing). Giving details of the deformation microstructure makes it easier to understand what happens during the subsequent annealing heat-treatment.

Reply #9> We appreciate the reviewer for exactly mentioning the critical points to be addressed about the precipitation mechanisms, which makes it easier to understand the microstructural evolution of the current alloys. We used two different meanings of shear bands in this study; macro-shear band and micro-shear band. According to the reviewer’s suggestion, we have added the details of the cold-rolled state in Supplementary Fig. 2. The macro-shear

bands are formed during cold-rolling and consequent strain localization, resulting in the thick lines distributed along ~10–45 degrees to the rolling direction, as shown in Supplementary Fig. 2b,c. As the reviewer mentioned, these macro-shear bands do not need to follow crystallographic orientations like slip systems. They play a significant role in producing fine recrystallized FCC grains during subsequent annealing processes as displayed in Fig. 2a.

On the other hand, inside FCC grains alongside the macro-shear bands, there are micro-shear bands which consist of mostly dislocation bands, which indicates that they are aligned along specific crystallographic planes. It is well known that in FCC-structured alloys micro-shear bands consist of dislocations, bundles of stacking faults, and twins which are developed along $\{111\}$ slip planes²⁹. In Supplementary Fig. 2d-f, the EBSD IPF, IQ, and KAM maps show the deformed FCC grains were subdivided into smaller grains by thick macro-shear bands as indicated by white arrows. Inside FCC grains near the macro-shear bands, the line features were developed exactly along one of $\{111\}$ plane traces, indicating that the bands well correspond to grain the crystallographic orientation. In addition, an ECCI image at higher magnification (Supplementary Fig. 2c) shows that the deformation structure in FCC grains is irrespective of macro-shear bands, indicating the development of micro-shear bands. These bands have a critical effect on generating intragranular $L2_1$ precipitates along $\{111\}$ plane traces inside non-recrystallized FCC grains, as shown in Fig. 2a-d.

Based on the above discussions, the following paragraph on page 5 has been revised to clarify ambiguous expressions about shear bands, and the following reference has been added in the manuscript. All terms of shear bands have been entirely revised as well as this paragraph.

- Figure 1c displays the overall schematics of the thermomechanical processing to fabricate the precipitation-strengthened \$Al_{0.2}CoNiV\$ alloys. As predicted, the homogenized state consisted of a coarse FCC matrix and Al-rich \$L2_1\$ islands

(Supplementary Fig. 1), in which the cold-rolling process induced macroscopic and microscopic shear bands (Supplementary Fig. 2). The macro-shear bands were formed during cold-rolling and consequent strain localization, resulting in thick lines distributed at ~10–45 degrees to the rolling direction. These macro-shear bands do not need to follow crystallographic orientations. On the other hand, micro-shear bands or called microbands²⁹ were developed inside the FCC grains alongside the macro-shear bands. The microbands consisted mostly of dislocation bands and thus they were aligned along specific crystallographic planes. After subsequent recrystallization and aging processes (1150 °C for 2 h and 850 °C for 1 h, respectively), referred to as RA, the L₂₁ phase was also observed in the form of islands and particles at grain boundaries or triple junctions (Supplementary Fig. 3). These L₂₁ precipitates at grain boundaries do not significantly contribute to strengthening because it is difficult for dislocations to directly interact with precipitates during glides, but they rather readily pile up in front of precipitates at the pre-existing pile-up sites, *i.e.*, the grain boundaries. Instead, here we utilized the abundant macro- and micro-shear bands as nucleation sites for precipitates and defect-free grain structures via recrystallization in addition to the grain boundaries. Therefore, the heat treatment conditions were selected to be 800 °C, 850 °C, and 900 °C for 1 h to control the recrystallization and precipitation behavior..

- **Supplementary Fig. 2 a** Tensile stress-strain curve of the cold-rolled state, **b, c** optical and SEM images showing macro-shear bands along ~10–45 degrees to the rolling direction, **d-f** EBSD IPF, IQ, and KAM maps showing macro-shear bands and micro-shear bands indicated by yellow and white arrows, respectively.
- **Reference 29.** Yoo, J. D. & Park, K.-T. Microband-induced plasticity in a high Mn–Al–C light steel. *Mater. Sci. Eng. A* **496**, 417-424 (2008).

Question #10> Line 138: “will be further discussed below”, this sentence may indicate that (K-S) orientation relationship is discussed immediately after this paragraph, but in fact it is discussed in a later part.

Reply #10> We appreciate for informing us know that we used an inappropriate phrase. We have modified the previous phrase to the following phrase on page 8.

- ... These L2₁ nanoparticles have Kurdjumov–Sachs (K–S) orientation relationship with FCC matrix, **which will be discussed later in this report.** ...

Question #11> Lines: 150-151: “L21 precipitates start to form at the grain boundaries or triple junctions ...” a fraction of L21 precipitates in A900 sample also formed inside the recrystallized grains as isolated precipitates or connected to annealing twin boundaries. This seems to be a characteristic of L21 phase and should not go unnoticed in this part.

Reply #11> We appreciate the helpful comment. It is correct that L2₁ precipitates form both at grain boundaries and inside the recrystallized grains in the A900 sample, as depicted in the schematic of microstructure in Fig. 2k. We have modified the following paragraph on page 9 to convey more accurate information to the readers. In addition, the fraction and size of the L2₁ precipitates inside grains have been measured and added in Supplementary Table 1 and 2.

- ... For the fully recrystallized sample (A900), the L2₁ particles at grain-boundaries or triple junctions in the recrystallized FCC region are ~320 nm in size. The L2₁ precipitates are also present inside the recrystallized FCC grains as intergranular particles. These intragranular L21 particles show an average size of ~146 nm. ...

Supplementary Table 1. Fractions of constituent phases for the Al_{0.2}CoNiV alloy annealed under three conditions (%).

Region		A800	A850	A900
L2 ₁ island	...			
Recrystallized	Total	33.9 ± 2.7	60.7 ± 3.5	90.4 ± 3.0
	FCC	27.8 ± 0.4	53.4 ± 0.9	83.9 ± 0.2
	L2 ₁ at GB*	5.6 ± 0.4	6.3 ± 0.5	5.4 ± 0.1
	L2 ₁ at IG**	0.5 ± 0.1	1.1 ± 0.3	1.1 ± 0.3
Non-Recrystallized	...			

*GB: Grain boundary, **IG: Inside grain, ***SB: Shear band

Supplementary Table 2. Sizes of FCC, L2₁, and σ phases for the Al_{0.2}CoNiV alloy annealed under three conditions.

Region	Phase	A800	A850	A900
L2 ₁ island	...			
Recrystallized FCC	FCC (μm)	1.1 \pm 0.7	1.7 \pm 0.9	2.4 \pm 1.7
	L2 ₁ at GB* (nm)	213.7 \pm 10.2	231.4 \pm 154.2	320.1 \pm 160.2
	L2 ₁ at IG** (nm)	90.1 \pm 2.7	143.0 \pm 6.2	146.4 \pm 6.3
Non-Recrystallized				

*GB: Grain boundary, **IG: Inside grain, ***SB: Shear band

Question #12> Lines 151-152: “L2₁ particles exhibit an incoherent or only one-sided semicoherent interface ... (Supplementary Fig. 4)”, assuming that an interface between L2₁ and FCC matrix follows (K-S) orientation relationship, does it necessarily mean that the interface is semicoherent?

Reply #12> We appreciate the opportunity to provide a clear explanation regarding the meaning of the semicoherent interface. The response to this suggestion is described in Reply #4. The K-S orientation relationship indicates two corresponding phases are aligned at specific directions on specific planes, forming complex semicoherent interfaces and thus lowering interfacial energy^{R1}. The direct evidence for the semi-coherent interface can be found in the HRTEM image of Fig. 3h in the original submission. To identify more accurately, we have added the magnified HRTEM image and corresponding inverse FFT image in Supplementary Fig. 6.

- **Supplementary Fig. 6 a** High-resolution TEM and corresponding FFT images, **b** inverse FFT image showing semi-coherent interface between γ and $L2_1$.
- **Reply Reference R1.** Dahmen, U. Orientation relationships in precipitation systems. *Acta Metall.* **30**, 63-73 (1982).

Question #13> Lines 164-165: "... evident green lines along $\{111\}$ plane trace ... along the shear bands.", why those green-line features in KAM map that appear along different $\{111\}$ plane traces are considered as shear bands?

Reply #13> We appreciate the helpful comment about the shear bands, which enables us to define the meaning of shear bands more precisely. Regarding two different meanings of shear bands in this study, *i.e.*, macro-shear band and micro-shear band, please refer to Reply #9. The green-line features indicate micro-shear bands. In KAM map of Fig. 3b, although not all green lines are straight but have a curvature, they are well aligned along $\{111\}$ plane traces. This curvature originates from the inhomogeneous and severe plastic deformation during cold-rolling. In addition, these KAM lines might become clearer as recovery and dislocation rearrangement along $\{111\}$ planes, *i.e.*, polygonization, were activated in non-recrystallized grains for the A800 and A850 alloys. Therefore, the following sentence has been modified on **page 9** to identify the shear bands more clearly.

- ... Compared to the dislocation-free recrystallized grains, the non-recrystallized FCC grains show evident green lines along {111} plane trace indicating a high dislocation density ($1.77 \times 10^{15} \text{ m}^{-2}$) of non-recrystallized grains. This result demonstrates that the cold-rolling leads to the micro-shear bands along specific crystallographic {111} planes as well as to macro-shear bands. ...

Question #14> Lines 188-193: “A proximity histogram across these two phases ... so-called Heusler phase.” I think using the proximity histogram is suitable when particles (or clusters) are fully embedded inside the APT tip. In Fig. 3i, it is better to provide a 1-D composition profile across a cylindrical ROI perpendicular to the interface and having at least 20nm length in each phase. That may be a better estimation for the elemental distribution in each phase. In addition, it seems that the measured composition of $L2_1$ (32V-32Co-18Ni-18Al) deviated from the stoichiometry of so-called Heusler phase $(\text{Co,Ni})_2\text{VAl}$. Is there any specific reason for that?

Reply #14> We appreciate for letting us know the correct data processing for APT analysis. According to the reviewer’s suggestion, we have changed the proximity histogram into a 1-D composition profile and corresponding concentration profiles in Fig. 3i. The estimated composition of $L2_1$ is 33V-33Co-18Ni-16Al, which is slightly different to the original composition. Regarding the deviation from the stoichiometry of the Heusler phase $(\text{Co,Ni})_2\text{VAl}$, our DFT calculations show that an enrichment in Co at the expense of Ni in the $L2_1$ phase is energetically more favorable, as demonstrated by the lower formation free energies at higher Co content (Fig. 1a) and justified by the "pseudogap" argument (Fig. 1b). For V and Al we could not reveal the origin in terms of formation free energies, but we believe that these deviations result from characteristics of high- and medium-entropy alloys presenting a high

degree of random mixing. We reserve the study of this feature for future investigation. The following sentences and figure have been added on page 5 and 11 in the manuscript.

- **Fig. 3 The microstructure evolution in the medium-entropy $\text{Al}_{0.2}\text{VCoNi}$ alloy heat-treated at 800 °C for 1 h. ... i APT tip reconstruction and 1-D profile across the L_{21} and FCC matrix. The phase boundary is highlighted by a 7.5 at% Al iso-concentration surface.**
- ... **As for the stoichiometry of this L_{21} phase, we note that the larger the Co concentration, the closer the pseudogap to E_F , signaling increased stability of the L_{21} phase with increasing amount of Co in this precipitate** (See Supplementary Note for detailed information). ...
- ... Figure 3i shows an APT reconstruction acquired from the non-recrystallized grain, which includes both FCC matrix and an L_{21} precipitate. **An 1-D concentration profile** across these two phases reveals that Al partitions to L_{21} , while V and Ni partition to FCC, and Co shows no pronounced partitioning. The measured chemical compositions

of the FCC matrix and $L2_1$ were 40V–34Co–24Ni–2Al (at%) and 33V–33Co–18Ni–16Al (at%), respectively. Consequently, the precipitates are identified as the $(\text{Co,Ni})_2\text{VAl}$ -type $L2_1$, so-called Heusler phase.

Question #15> Fig. 4 and the assessment of mechanical properties: it is clear that VCoNiAl_{0.2} shows an excellent combination of strength-ductility, but overall it does not seem that the alloy containing semicoherent precipitates can overcome the strength-ductility trade-off. In particular, the ductility of A800 sample having the highest fraction of semicoherent precipitates dropped substantially compared to A850 and A900. In this regard, semicoherent precipitates seem to have a similar degrading effect on ductility as a TCP phase does (e.g. σ phase). In Fig. 4b-c, the author made a nice effort to compare the mechanical properties of VCoNiAl_{0.2} with other references. As an additional reference, I recommend adding the data associated with the single-phase VCoNi system that was previously studied by the authors (Ref. 14). Here, the critical question is that whether the strength-ductility combination in the presence of semicoherent precipitates (VCoNiAl_{0.2}) surpasses that of single-phase VCoNi alloy or not.

Reply #15> We thank the reviewer for this comment, which gives us the opportunity to elaborate on the mechanical properties of the Al_{0.2}CoNiV alloys. As the reviewer suggested, we have added the mechanical properties of single-phase CoNiV alloys to Fig. 4b and Fig. 4c for comparison with the present alloys. As shown in Fig. 4b-c, it is correct that the Al_{0.2}CoNiV follows a linear trend of the strength-ductility combination of the CoNiV alloys. Therefore, the overstated expression regarding overcoming the trade-off has been removed in the revised manuscript.

However, in the case of the CoNiV alloy, it should be noted that an approximately 1 GPa yield strength was achieved for the alloy of 2 μm in average grain size, which is close to the

minimum size acquirable in conventional thermomechanical processing. In other words, there is a limitation to enhance grain boundary strengthening by further refinement of the grain size. In addition, the single FCC phase in CoNiV alloy partly decomposes to κ phase below 850 °C and entirely to κ and σ phases below 800 °C, which cause severe brittleness³¹. This limited process windows restrict the CoNiV to possess finer grain size less than $\sim 2 \mu\text{m}$. This challenge implies that another strengthening mechanism should be embodied in this alloy to further improve the mechanical property.

In this respect, to make ultrastrong ductile CoNiV-based MEAs with yield strength over 1 GPa, we added Al to suppress the embrittling phases and utilize the L_{21} phase, and we introduced the abundant nucleation sites to generating uniformly distributed nanosized precipitates. Consequently, we developed the $\text{Al}_{0.2}\text{CoNiV}$ alloys with yield strength and ductility of $\sim 1262 \text{ MPa}$ and 26.2 %, respectively, in the optimal processing condition. As well as the yield strength improvement, the L_{21} phase has a favorable role in sustaining a high strain-hardening rate in gigapascal-grade high strength, as described in Reply #6. In accordance with the responses, the following figures and sentences have been revised on page 12, and the following reference has been added in the manuscript.

- **Fig. 4 Room-temperature mechanical properties of our alloys.** ... **b** Overview of yield strength versus uniform elongation, and **c** **Ultimate** tensile strength versus total

elongation values for the current $\text{Al}_{0.2}\text{CoNiV}$ alloy, compared to single or multiphase high-/medium-entropy alloys. ...

- ... The alloys designed in this work exhibit an excellent combination of ultrahigh strength and uniform ductility, surpassing the previously reported HEAs and MEAs having semicoherent and incoherent phases, and being comparable to the recently reported ultrastrong and ductile coherent L_{12} precipitation-strengthened HEAs and MEAs. The mechanical properties of the present alloys seem to follow a linear trend of the strength-ductility combination of the CoNiV alloys. It should be noted that it is challenging further enhance the mechanical properties of CoNiV alloys, as the limited process windows in conventional processing restrict the refinement of grains to sizes smaller than $\sim 2 \mu\text{m}^3$. Another strengthening mechanism should be embodied and the present approach enables further improvement in mechanical properties.
- **Reference 31.** Sohn, S. S. *et al.* High-rate superplasticity in an equiatomic medium-entropy VCoNi alloy enabled through dynamic recrystallization of a duplex microstructure of ordered phases. *Acta Mater.* **194**, 106-117 (2020).

Question #16> Lines 239-245: it may not be a good idea to generally discuss a list of defects (dislocation, dislocation walls, NTs, SFs) that can possibly be a nucleation site for precipitation. Aside from that, all those defects mentioned above can form along $\{111\}$ planes without the presence/contribution of shear bands. Is it really necessary to relate the underlying precipitation mechanism to the so-called shear-band driven precipitation?

Reply #16> We appreciate for indicating the critical point related to the precipitation mechanisms. First, as the reviewer suggested, a major contribution results from dislocations

and partly from NTs and SFs which were observed near L2₁ particles in TEM images. Thus, the following unnecessary statement has been removed:

- **Fig. 3 The microstructure evolution in the medium-entropy Al_{0.2}CoNiV alloy heat-treated at 800 °C for 1 h. ... g, h** High-resolution TEM and corresponding FFT images indicate the **micro**-shear band consists of stacking faults and nanotwins, **providing nucleation sites for the L2₁ phase.**
- ... High-resolution TEM and fast Fourier-transform (FFT) images confirm that the **micro**-shear bands consist of stacking faults (SFs) and nanotwin (NT) bundles (Fig. 3g, h), **indicating that shear micro bands act as nucleation sites for the L2₁ phase.** ...
- ... **In addition to dislocations, numerous SFs and NTs along the shear bands in non-recrystallized FCC grains provide abundant nucleation sites for L2₁ precipitates (Fig. 3e-h).** ...

Second, we are positive about the relation between the underlying precipitation mechanism and the shear-band driven precipitation. This misleading might result from our ambiguous use of shear bands in the original manuscript; the classification details of macro- and micro-shear bands are described in Reply #9. The precipitates evidently align along {111} plane traces of non-recrystallized FCC grains having K-S orientation relationship, as shown in ECCI and EBSD micrographs. Therefore, the micro-shear bands play a critical role in providing nucleation sites for L2₁ precipitates.

Question #17> Lines 246-247: “These underlying mechanisms ... are confirmed in Supplementary Fig. 4”, This is a generalized statement. I don’t think Fig. 4 alone reveals any underlying mechanisms.

Reply #17> We appreciate the helpful comment. After revising the sentence carefully, we regarded it unnecessary and decided to remove the sentence.

- ~~These underlying mechanisms for forming nanosized intragranular L2₁ particles are confirmed in Supplementary Fig. 4...~~

Question #18> Lines 253-258: as the authors discussed, the smaller size of intragranular L2₁ in A800 can be related to their semicoherent interfaces (low mobility). However, I think another important factor is the location of precipitation in A800. The fine intragranular precipitates in A800 formed inside the non-recrystallized grains (far from GBs). In contrast, in A900 with recrystallized microstructure, intergranular precipitates had direct access to the GBs that could act as easy diffusion paths assisting the rapid growth of intergranular precipitates. For comparison, there were some in-grain precipitates in A900 (Supp. Fig. 4b), which were clearly smaller than the intergranular precipitates, indicating that access to the GB is crucial for the precipitate growth.

Reply #18> We appreciate for presenting valuable insight into our discussion. Indeed, in the A900 alloy, the intergranular precipitate is large due to easy diffusion at GBs, while the intragranular precipitate is smaller than the other. We entirely agree with the comments, thus the following sentences have been added on page 14.

- ... **In addition to the difference in the type of interfaces, the location of the precipitates contributes to their size difference. As shown in Supplementary Fig. 5b, the precipitates at grain boundaries in the A900 alloy with a recrystallized microstructure have a larger size than the precipitates inside the grains. The intergranular precipitates have direct access to the grain boundaries that could act as easy diffusion paths assisting the rapid growth of intergranular precipitates.** Accordingly, Intragranular L2₁ precipitates along

the shear bands become finer with an average size of ~60 nm compared to intergranular precipitates at the grain boundaries, ...

Question #19> Lines 261-272: the authors put efforts into estimating the contribution of each strengthening mechanism separately, although the methods and calculations they used and described in Supp. Data seem to be oversimplified. But assuming that such estimation and the corresponding Fig. 5 (Supp. Data) are accurate enough, this means that in A850 having the best combination of strength-ductility, only ~10% of strength originates from the semicoherent precipitates in non-recrystallized regions. If so, the strategy of using semicoherent-precipitation strengthening becomes a minor factor, and authors are not able to claim that the excellent properties of A850 are due to their proposed strategy.

Reply #19> We appreciate the reviewer's helpful comment to improve the originality of the present study with respect to the role of L2₁ precipitate. We agree with the reviewer's concern. For the A850 alloy, the strengthening contributions from solid-solution, grain boundary, dislocation, and precipitation strengthening were calculated to be 383 MPa, 478 MPa, 138 MPa, and 126 MPa, respectively. Thus, the precipitation strengthening contributes approximately ~11% to the calculated yield strength (1125 MPa).

However, this quantity (or 1262 MPa for A850) is hard to be achieved for the CoNiV alloy having single FCC structure. As mentioned in Reply #15, although the CoNiV alloy shows an approximately 1 GPa yield strength at 2 μm in average grain size, the limited process windows prevent to achieve finer grain size less than ~2 μm . This limitation implies that additional strengthening mechanism must be implemented to further improve mechanical properties. Therefore, the addition of Al to suppress the embrittling phases and utilize L2₁ phase, combined with the abundant nucleation sites to generate a uniform distribution of nanosized precipitates,

resulted in the $\text{Al}_{0.2}\text{CoNiV}$ alloys with yield strength and ductility of ~ 1262 MPa and 26.2 %, respectively, in the optimal processing condition.

As well as the yield strength improvement, the L2_1 phase has a critical role on strain-hardening mechanism in gigapascal-grade high strength, as described in Reply #6 and #15. As a result of additional STEM observations in this revision, we have found that the dislocations pile-up and consequent reduction of mean free path of dislocations are promoted in addition to the conventional Orowan mechanism. Combined with the formation of dislocation substructures in the base CoNiV alloy, this further mean free path refinement effect could lead to the considerable strain-hardening effect. Therefore, a large uniform elongation was achieved at very high stress level.

In conclusion, the present strategy of using semicoherent L2_1 precipitates does overcome the limitations of the CoNiV alloy, improves the yield strength by precipitation hardening, and contributes to the strain-hardening rate considerably. Therefore, we believe the proposed strategy played a dominant role in resulting in the excellent properties of the A850 alloy.

Question #20> Line 284-285: “the A800 sample exhibits low ductility due to the initially high dislocation density and the limited fraction of the dislocation-free recrystallized grains”, this is a reasonable statement, but the degrading effect of precipitates on ductility cannot be simply ignored. Have authors conducted any fracture surface analysis? I understand carrying out experiments during the COVID-19 pandemic can be difficult, but it would be interesting to see whether the fracture mechanism in this sample is related to the semicoherent particles or not.

Reply #20> We appreciate for the helpful suggestion. We have conducted fracture surface analyses to find out the role of semicoherent particles on low tensile ductility for the A800 alloy. The SEM observations reveal that only ductile dimple fracture occurred for all three

alloys. We guess the σ phase in the A800 alloy might induce the embrittlement, but the brittle fracture appearance was not observed in the entire area. This result indicates the σ and $L2_1$ phases have little effect on the brittle crack initiation or propagation, whether located inside the grains or at the grain boundaries. Accordingly, we could attribute the low ductility of the A800 specimen to the high fraction of the non-recrystallized region with initially high dislocation density. The fractography has been added in Supplementary Fig. 7, and the relevant explanations have been added on page 16.

- ... Therefore, the A800 sample exhibits low ductility due to the initially high dislocation density and the limited fraction of the dislocation-free recrystallized grains.

The fracture surface observations support the conclusion that the $L2_1$ and σ phases do not induce brittle characteristics (Supplementary Fig. 7). ...

- **Supplementary Fig. 7** Tensile fractographies of the **a** A800, **b** A850, and **c** A900 alloys. Only ductile-dimple fracture was observed.

Other comments:

Question #21> In the introduction part, the authors described the advantage of using coherent precipitates over incoherent precipitates (or TCP phases). Then they designed an alloy having semicoherent precipitates and examined the microstructures and mechanical properties. But after looking at the results and discussion, it is not yet convincing whether there is a big

advantage of using semicoherent phases over incoherent phases. However, I do think that the semicoherent L21 in VCoNiAl0.2 has a unique behavior, i.e. the easy/profound nucleation of L21 along {111} planes containing deformation-induced defects, which cannot be commonly seen in other types of precipitate-containing alloys. I think the authors can elaborate on this fact to improve the discussion part.

Reply #21> We appreciate the reviewer's positive response to our strategy. Thanks to the comments, we could elaborate on this fact to improve the discussion part in terms of nucleation behaviors of the intragranular precipitates. Accompanying the nucleation behavior, the semicoherent interface has lower interfacial energy than the incoherent interface, leading to a slower growth rate in the precipitates. The studies related to the interfacial energy would be worth investigations in research fields where the control of the interfacial energy of the precipitates is important (e.g., hydrogen embrittlement, radiation). The following sentence has been added on page 14.

- ... Thus, the easy and profound nucleation of the L2₁ phase is achieved along {111} planes containing deformation-induced defects, which cannot be commonly seen in other types of precipitate-containing alloys.

Question #22> Phase identification of VCoNiAl0.2 alloy via XRD analysis is quite important (evidence for L21 phase). I recommend showing XRD profile as a main figure in the manuscript, not as supplementary data.

Reply #22> We appreciate for constructive comment. We have moved the XRD profile into Fig. 1d and modified the relevant explanation on page 7 in the manuscript as follows.

- **Fig. 1 Material and process design. ... d Phase identification via XRD analysis of the annealed Al_{0.2}CoNiV alloys.**
- ... **As shown by X-ray diffraction (XRD) (Fig. 1d), the primary peaks of the alloys corresponded to the FCC phase, while the secondary peaks were superlattice reflections stemming from the L₂₁ ordered phase. The A850 and A800 alloys additionally contained the σ phase, present for the most part in the L₂₁ islands and with a smaller fraction also in the nearby recrystallized FCC grains (Supplementary Fig. 4). Figure 2a–f shows electron backscatter diffraction (EBSD) maps for the investigated alloys, revealing FCC (Fig. 2a–c) and L₂₁ (Fig. 2d–f) phases. ...**

Question #23> Some detailed information needs to be added to some figure captions: for example, red lines in Fig. 4 (Supp. Data) indicates semicoherent interfaces or “HA” in Fig. 4 stands for “homogenized & aged”, or symbols σ PH, etc. in Fig. 5 (Supp. Data) represent ... It is much easier to understand a figure as it stands without searching for necessary information in the text.

Reply #23> We appreciate for kindly indicating details to be improved. As the reviewer recommended, we have added the detailed information to each figure as follows:

- **Fig. 4 Room-temperature mechanical properties of our alloys.** a Engineering tensile stress–strain curves for the annealed Al_{0.2}CoNiV alloys. ...
- **Supplementary Fig. 5** EBSD IQ maps of annealed Al_{0.2}CoNiV alloys superimposed by phase color. ...
- **Supplementary Fig. 8** Summarized chart showing calculated strengthening contributions from each mechanism for alloys annealed at 800, 850, and 900 °C for 1 h.

Reviewer #3 (Remarks to the Author):

The paper “Shear band-driven precipitate dispersion for ultrastrong ductile medium-entropy alloys” presents characterization of a complex concentrated alloy Al_{0.2}CoNiV. Tensile deformation testing of the alloy revealed very high yield strength of 1260 MPa in combination with sufficient ductility of 27%. This is definitely an interesting finding. High strength of the alloy is attributed to the precipitation strengthening by finely dispersed particles of L21 Heusler phase. These particles have well defined orientation relationship with the disordered fcc matrix and form semicoherent interfaces with the matrix. Although the results reported in the paper, in particular results of the microstructure investigations, are interesting there are some points which have to be improved and/or explained better.

Question #24> According to the convention used for designating of complex concentrated alloys the constituting elements should be mentioned in the alphabetic order. For the present alloy it reads Al_{0.2}CoNiV.

Reply #24> We appreciate the remark on the notation of complex concentrated alloys. We have modified the nomenclature to follow the alphabetical order, *i.e.*, VCoNi to **CoNiV**, and VCoNiAl_{0.2} to **Al_{0.2}CoNiV** throughout the manuscript.

Question #25> It is surprising that the authors do not present results of fractography, *i.e.* surfaces of samples broken in the tensile deformation test, in order to analyze the mode of deformation of Al_{0.2}CoNiV alloy and to examine whether it exhibits transgranular or intergranular fracture.

Reply #25> We sincerely thank you for giving advice on results we haven't considered. The response to this suggestion has been described in Reply #20. To sum up, the SEM observations

reveal that only ductile dimple fracture occurred for all three alloys. We guess that the σ phase in the A800 alloy might induce the embrittlement, but the brittle fracture appearance was not observed in the entire area. This result indicates the σ and L21 phases have little effect on the brittle crack initiation or propagation, whether located inside the grains or at the grain boundaries. The fractographies have been added in Supplementary Fig. 7.

- **Supplementary Fig. 7** Tensile fractographies of the **a** A800, **b** A850, and **c** A900 alloys. Only ductile-dimple fracture was observed.

Question #26> Recrystallized grains contain a lot of annealing twins visible in Fig. 2i and shear bands with L21 particles contain nanotwins. However it seems that contribution of twins is not included in the strengthening model presented in lines 348-414.

Reply #26> We appreciate the helpful comment. As the reviewer suggested, two types of twins are present in the $\text{Al}_{0.2}\text{CoNiV}$ alloys; annealing twins in recrystallized region and deformation nanotwins in non-recrystallized region. Both types can contribute to strengthening by acting as a barrier for dislocation glide and reduce dislocation mean free path. In terms of the annealing twins, their strengthening contribution is already included in grain-boundary strengthening ($\Delta\sigma_{GB}$). As the average grain size is measured from EBSD analysis, the annealing twins are classified as different grains from their primary grain because they have a different crystallographic orientation compared to the primary grains. This approach is a general

treatment to deal with annealing twins in FCC-structured alloys. Therefore, the strengthening contribution of the annealing twins is contained in the Hall-Petch relationship in terms of the average grain size.

In contrast to the annealing twins, whose strengthening contribution can be readily quantified by grain-boundary strengthening, it is very difficult to quantify the strengthening contribution of the deformation nanotwins in present alloys. They have a very fine width of a few nanometers and can only be observed in a nanometer-scale investigation such as TEM. We think it is not reasonable to account their strengthening contribution of the alloys from the finite range of TEM observations. In addition, as shown in Supplementary Fig. 9, the major plastic deformation mechanism of the $\text{Al}_{0.2}\text{CoNiV}$ alloys is the dislocation slip. Therefore, the strengthening contribution of the deformation nanotwins is considered negligible compared to that of the high dislocation density in the non-recrystallized region.

Question #27> It is mentioned in the manuscript (lines 153-154) that the L2_1 particles occur at grain-boundaries or triple junctions in the recrystallized fcc region. But in Fig. 2i the L2_1 particles can be seen not only in grain boundaries but also inside grains.

Reply #27> We appreciate the helpful comment. It is correct that L2_1 precipitates form both at grain boundaries and inside the recrystallized grains in the A900 sample, as depicted in schematic of microstructure in Fig. 2k. We have modified the following paragraph on page 9 to convey more accurate information to the readers. In addition, fraction and size of the L2_1 precipitates inside grains have been measured and added in Supplementary Table 1 and 2.

- ... For the fully recrystallized sample (A900), the L2_1 particles at grain-boundaries or triple junctions in the recrystallized FCC region are ~320 nm in size. The \$\text{L2}_1\$

precipitates are also present inside the recrystallized FCC grains as intergranular particles. These intragranular L21 particles show an average size of ~146 nm. ...

Supplementary Table 1. Fractions of constituent phases for the Al_{0.2}CoNiV alloy annealed under three conditions (%).

Region		A800	A850	A900
L2 ₁ island	...			
Recrystallized	Total	33.9 ± 2.7	60.7 ± 3.5	90.4 ± 3.0
	FCC	27.8 ± 0.4	53.4 ± 0.9	83.9 ± 0.2
	L2 ₁ at GB*	5.6 ± 0.4	6.3 ± 0.5	5.4 ± 0.1
	L2 ₁ at IG**	0.5 ± 0.1	1.1 ± 0.3	1.1 ± 0.3
Non-Recrystallized	...			

*GB: Grain boundary, **IG: Inside grain, ***SB: Shear band

Supplementary Table 2. Sizes of FCC, L2₁, and σ phases for the Al_{0.2}CoNiV alloy annealed under three conditions.

Region	Phase	A800	A850	A900
L2 ₁ island	...			
Recrystallized FCC	FCC (μm)	1.1 ± 0.7	1.7 ± 0.9	2.4 ± 1.7
	L2 ₁ at GB* (nm)	213.7 ± 10.2	231.4 ± 154.2	320.1 ± 160.2
	L2 ₁ at IG** (nm)	90.1 ± 2.7	143.0 ± 6.2	146.4 ± 6.3
Non-Recrystallized				

*GB: Grain boundary, **IG: Inside grain, ***SB: Shear band

Question #28> Fig. 4: What is the difference between uniform elongation and total elongation?

Reply #28> Thank you for the interesting question. The elongation is the ratio of the change of gauge length to the initial gauge length during uniaxial tensile test, which represents the ductility of the materials. For sufficiently ductile materials, the elongation can be categorized

as uniform elongation, post-uniform elongation, and total elongation. The uniform elongation indicates the elongation until plastic instability occurs, called necking. Before necking, the specimen deforms uniformly in the gauge section and thus a reduction in cross-sectional area is uniform.

While, after the plastic instability occurs, the applied stress is concentrated in the necking region, and the deformation is consistently localized. The elongation from necking to fracture is called post-uniform or post-necking elongation. Finally, the total elongation is the elongation when fracture occurs and includes uniform and post-uniform elongations. The uniform elongation better represents the ductility of materials in uniaxial deformation as the post-uniform elongation depends on the geometry of specimens. However, for some materials that do not show necking, such as our A800 alloy, the total elongation can be a sole parameter to indicate the ductility of materials.

Question #29> In the text of the manuscript and also in the Supplementary Table 3 the authors use the term “tensile strength” but actually they probably mean “ultimate tensile strength”. This should be corrected everywhere in the paper.

Reply #29> We appreciate for helpful comment. It is correct to use the term “ultimate tensile strength” instead of “tensile strength”. We corrected “tensile strength” to “ultimate tensile strength” everywhere in the paper.

Question #30> Experimental uncertainties should be mentioned in the relative concentrations of phases in Supplementary Table I.

Reply #30> Thank you for indicating experimental uncertainties that we haven't considered significantly. We have performed additional EBSD and ECCI analyses for each sample and summarized them in Supplementary Table.1. All quantities are rounded to one decimal point.

Supplementary Table 1. Fractions of constituent phases for the $\text{Al}_{0.2}\text{CoNiV}$ alloy annealed under three conditions (%).

Region		A800	A850	A900
L2 ₁ island	Total	11.3 ± 2.6	9.6 ± 3.7	9.6 ± 3.0
	FCC	5.5 ± 0.5	4.0 ± 0.2	2.8 ± 0.3
	L2 ₁	4.8 ± 0.4	5.2 ± 0.2	6.8 ± 0.3
		1.0 ± 0.1	0.4 ± 0.1	0
Recrystallized	Total	33.9 ± 2.7	60.7 ± 3.5	90.4 ± 3.0
	FCC	27.8 ± 0.4	53.4 ± 0.9	83.9 ± 0.2
	L2 ₁ at GB*	5.6 ± 0.4	6.3 ± 0.5	5.4 ± 0.1
	L2 ₁ at IG**	0.5 ± 0.1	1.1 ± 0.3	1.1 ± 0.3
Non-Recrystallized	Total	54.8 ± 5.0	29.7 ± 4.0	0
	FCC	46.0 ± 1.3	25.3 ± 0.2	0
	L2 ₁ at SB***	8.8 ± 1.3	4.40 ± 0.2	0

*GB: Grain boundary, **IG: Inside grain, ***SB: Shear band

Question #31> All uncertainties in Supplementary Table 2 and Table 3 have to be rounded to one standard digit and measured data have to be rounded to the order of the last significant digit of the uncertainty.

Reply #31> We appreciate for your kind feedback regarding all uncertainties. The responses have been revised in the following Supplementary Table 2 and Table 3.

Supplementary Table 2. Sizes of FCC, L2₁, and σ phases for the $\text{Al}_{0.2}\text{CoNiV}$ alloy annealed under three conditions.

Region	Phase	A800	A850	A900
L ₂₁ island	L ₂₁ (μm)	7.4 ± 4.1	9.5 ± 7.6	8.4 ± 5.9
	FCC (nm)	40.1 ± 15.3	205.1 ± 85.7	250.4 ± 140.5
	(nm)	123.4 ± 82.0	105.1 ± 79.2	
Recrystallized FCC	FCC (μm)	1.1 ± 0.7	1.7 ± 0.9	2.4 ± 1.7
	L ₂₁ at GB* (nm)	213.7 ± 10.2	231.4 ± 154.2	320.1 ± 160.2
	L ₂₁ at IG** (nm)	90.1 ± 2.7	143.0 ± 6.2	146.4 ± 6.3
Non-Recrystallized	FCC (μm)	212.1 ± 73.4	91.3 ± 18.7	
	L ₂₁ at SB*** (nm)	57.3 ± 7.4	86.6 ± 34.6	

*GB: Grain boundary, **IG: Inside grain, ***SB: Shear band

Supplementary Table 3. Room-temperature tensile properties of the Al_{0.2}CoNiV alloys.

Specimen	Yield Strength (MPa)	Tensile Strength (MPa)	Elongation (%)
A800	1500.3 ± 14.7	1727.1 ± 21.7	8.2 ± 1.2
A850	1262.1 ± 11.1	1586.6 ± 9.9	26.7 ± 3.1
A900	1050.4 ± 20.4	1479.8 ± 11.4	31.7 ± 4.4
RA	569.7 ± 8.2	1122.4 ± 11.3	48.7 ± 1.8

REVIEWERS' COMMENTS

Reviewer #1 (Remarks to the Author):

The authors have address most of the issues this reviewer raised. I recommend accepting this paper.

Reviewer #2 (Remarks to the Author):

In general, the authors well responded to the questions and comments. The additional figures and data in the revised version clarify the important points that were missing in the original manuscript, such as adding cold-rolled microstructure before heat-treatment, comparing mechanical properties between new alloys and the base CoNiV alloy, discussing deformation structure after tensile test, fracture surface study, etc. The figures are informative and well labeled in the revised version.

- Regarding the assessment of mechanical properties in Fig.4:

After adding the mechanical properties of the single-phase CoNiV to Fig. 4b/c, it becomes clear that the newly designed Al0.2CoNiV alloy follows a similar linear trend of the CoNiV. This indicates that the original claim of overcoming strength-ductility trade-off was somewhat overstated, which is reconsidered and corrected by the authors in the revised version.

Nevertheless, there is no doubt the new alloy has an excellent combination of strength-ductility. In particular, the authors provide new information describing that the limitations imposed on the grain refinement strengthening in CoNiV were another reason to design the new alloys. This is a quite important point and better to be added (at least mentioned briefly) in the Introduction part.

- Regarding using the term “micro-shear band”:

First, it is helpful that the authors specified macro-shear bands and micro-shear bands, where only the latter seems to play the key role in the nucleation of nanosized/finely dispersed precipitates. Second, I do not deny the correlation between the underlying precipitation mechanism and micro-shear banding. But if I were the author, I would be more cautious to use the term micro-shear band, only based on the linear features that appear along {111} plane traces in SEM or EBSD images, as it is claimed in Supp. Fig.2. Linear features could be micro-shear bands, microbands (related but different from micro-shear bands), twin bundles, or linear arrays of dislocations. Those terms have specific definitions and should not be arbitrarily used.

Reviewer #3 (Remarks to the Author):

I appreciate that the authors substantially improved the quality of their manuscript and satisfactorily clarified most of the comments.

One remark regarding Question #31, by definition significant digit is any digit except of zeros before the

first significant digit. For example 14.7 has 3 significant digits (not one). So rounding to one significant digit does not mean rounding to tenths. The point is that uncertainties mentioned in the table are estimates obtained from several repeated measurements. Relative precision of this estimation is not better than 10%. It means that for example the uncertainty of the Yield strength of A800 specimen 14.7 MPa is definitely not known with the precision of 0.1 MPa. The error should be rounded to one significant digit, i.e. 10 MPa in the present case, and the Yield strength value should be rounded to the order of the last significant digit of the error, i.e. in state of (1500.3 ± 14.7) MPa the authors should write (1500 ± 10) MPa and so on for all values in Tables 2,3. It has really no sense to present data with high number of significant digits when these significant digits do not correspond to anything real but represent numerical noise only.

Reviewer #1 (Remarks to the Author):

The authors have address most of the issues this reviewer raised. I recommend accepting this paper.

Reply> We appreciate your positive opinion on our responses.

Reviewer #2 (Remarks to the Author):

In general, the authors well responded to the questions and comments. The additional figures and data in the revised version clarify the important points that were missing in the original manuscript, such as adding cold-rolled microstructure before heat-treatment, comparing mechanical properties between new alloys and the base CoNiV alloy, discussing deformation structure after tensile test, fracture surface study, etc. The figures are informative and well labeled in the revised version.

Remark #1> Regarding the assessment of mechanical properties in Fig.4:

After adding the mechanical properties of the single-phase CoNiV to Fig. 4b/c, it becomes clear that the newly designed Al_{0.2}CoNiV alloy follows a similar linear trend of the CoNiV. This indicates that the original claim of overcoming strength-ductility trade-off was somewhat overstated, which is reconsidered and corrected by the authors in the revised version. Nevertheless, there is no doubt the new alloy has an excellent combination of strength-ductility. In particular, the authors provide new information describing that the limitations imposed on the grain refinement strengthening in CoNiV were another reason to design the new alloys. This is a quite important point and better to be added (at least mentioned briefly) in the Introduction part.

Reply #1> We appreciate the helpful comment for better describing the motivation of our study.

Following sentences have been added in the Introduction part.

- ... This property is ascribed from only solid solution and grain boundary strengthening at an average $\sim 2 \mu\text{m}$ grain size in FCC-structured matrix. However, it is challenging to further enhance the mechanical properties of CoNiV alloys as an additional refinement of grains is restricted in conventional processing due to the limited process windows¹⁶. In this respect, precipitation strengthening can be an attractive candidate for further improving the mechanical properties. ...

Remark #2> Regarding using the term “micro-shear band”:

First, it is helpful that the authors specified macro-shear bands and micro-shear bands, where only the latter seems to play the key role in the nucleation of nanosized/finely dispersed precipitates.

Second, I do not deny the correlation between the underlying precipitation mechanism and micro-shear banding. But if I were the author, I would be more cautious to use the term micro-shear band, only based on the linear features that appear along $\{111\}$ plane traces in SEM or EBSD images, as it is claimed in Supp. Fig.2. Linear features could be micro-shear bands, microbands (related but different from micro-shear bands), twin bundles, or linear arrays of dislocations. Those terms have specific definitions and should not be arbitrarily used.

Reply #2> We appreciate the helpful comments. For the first suggestion, we have revised the following sentences in the 2nd paragraph of the Results part and in the 3rd paragraph of the Discussion part.

- ... Instead, here we utilized the abundant micro-shear bands as nucleation sites via partial recrystallization treatment for obtaining nanosized precipitates finely dispersed inside grains. ...
- ... Therefore, micro-shear bands play a key role in the nucleation of nanosized precipitates finely dispersed inside non-recrystallized grains, while macro-shear bands promote the formation of fine recrystallized grains and intergranular L2₁ particles.

For the second suggestion, we have revised the term ‘microbands’ as ‘micro-shear bands’ to avoid misleading. We agree that each terminology such as micro-shear bands, microbands, twin bundles, or linear arrays of dislocations has a specific definition. However, the term ‘shear bands’ can comprise the various defects occurring in slip planes (shear planes) in the form of ϵ (hcp) martensite, mechanical twins, dense stacking-fault bundles, or dislocations according to the following literature:

- 1) Murr, L. E., Staudhammer, K. P. & Hecker, S. S. Effects of Strain State and Strain Rate on Deformation-Induced Transformation in 304 Stainless Steel: Part II. Microstructural Study. *Metall. Trans. A* **13**, 627-635 (1982).
- 2) Choi, J.Y. & Jin, W. Strain induced martensite formation and its effect on strain hardening behavior in the cold drawn 304 austenitic stainless steels. *Scr. Mater.* **36**, 99-104 (1997).
- 3) Olson, G. & Cohen, M. Kinetics of strain-induced martensitic nucleation. *Metall. Trans. A* **6**, 791 (1975).

In addition, TEM results (Fig. 3f-h) showed that the micro-shear bands consist of stacking faults and nanotwin bundles although they seem not to play a dominant role. Therefore, to specify the micro-shear bands more clearly, the following sentence has been revised in the 2nd paragraph of the Results part and the related reference have been added.

- ... It is known that the micro-shear bands can be in the form of twins bundles, dense stacking-fault bundles, or linear arrays of dislocations and thus they are aligned along specific crystallographic shear planes³⁰. ...
- Reference 30. Olson, G. & Cohen, M. Kinetics of strain-induced martensitic nucleation. *Metall. Trans. A* **6**, 791 (1975).

Reviewer #3 (Remarks to the Author):

I appreciate that the authors substantially improved the quality of their manuscript and satisfactorily clarified most of the comments.

Remark #3> One remark regarding Question #31, by definition significant digit is any digit except of zeros before the first significant digit. For example 14.7 has 3 significant digits (not one). So rounding to one significant digit does not mean rounding to tenths. The point is that uncertainties mentioned in the table are estimates obtained from several repeated measurements. Relative precision of this estimation is not better than 10%. It means that for example the uncertainty of the Yield strength of A800 specimen 14.7 MPa is definitely not know with the precision of 0.1 MPa. The error should be rounded to one significant digit, i.e. 10 MPa in the present case, and the Yield strength value should be rounded to the order of the last significant digit of the error, i.e. in state of (1500.3 +/- 14.7) MPa the authors should write (1500 +/- 10) MPa and so on for all values in Tables 2,3. It is has really no sense to present data with high number of significant digits when these significant digits do not correspond to anything real but represent numerical noise only.

Reply #3> We appreciate the helpful comment for your kind feedback regarding all uncertainties. The responses have been revised in the following Supplementary Table 2 and Table 3.

Supplementary Table 2. Sizes of FCC, L2₁, and σ phases for the Al_{0.2}CoNiV alloy annealed under three conditions.

Region	Phase	A800	A850	A900
L2 ₁ island	L2 ₁ (μm)	7 ± 4	10 ± 8	8 ± 6
	FCC (nm)	40 ± 20	200 ± 90	300 ± 100
	σ (nm)	120 ± 80	110 ± 80	
Recrystallized FCC	FCC (μm)	1.1 ± 0.7	1.7 ± 0.9	2 ± 2
	L2 ₁ at GB* (nm)	210 ± 10	200 ± 200	300 ± 200
	L2 ₁ at IG** (nm)	90 ± 3	143 ± 6	146 ± 6
Non-Recrystallized	FCC (μm)	210 ± 70	90 ± 20	
	L2 ₁ at SB*** (nm)	57 ± 7	90 ± 30	

*GB: Grain boundary, **IG: Inside grain, ***SB: Shear band

Supplementary Table 3. Room-temperature tensile properties of the Al_{0.2}CoNiV alloys.

Specimen	Yield Strength (MPa)	Tensile Strength (MPa)	Elongation (%)
A800	1500 ± 10	1730 ± 20	8 ± 1
A850	1260 ± 10	1590 ± 10	27 ± 3
A900	1050 ± 20	1480 ± 10	32 ± 4
RA	570 ± 8	1120 ± 10	49 ± 2